# First quantitative high-throughput screen in zebrafish identifies novel pathways for increasing pancreatic β-cell mass

Guangliang Wang[1,2], Surendra K Rajpurohit[3], Fabien Delaspre[1,2], Steven L Walker[3], David T White[3], Alexis Ceasrine[4], Rejji Kuruvilla[4], Ruo-jing Li[5], Joong S Shim[6], Jun O Liu[5,7], Michael J Parsons[1,2]*[†], Jeff S Mumm[1,3]*[†][‡]

[1]McKusick-Nathans Institute of Genetic Medicine, Johns Hopkins University, Baltimore, United States; [2]Department of Surgery, Johns Hopkins University, Baltimore, United States; [3]Department of Cellular Biology and Anatomy, Georgia Regents University, Augusta, United States; [4]Department of Biology, Johns Hopkins University, Baltimore, United States; [5]Pharmacology and Molecular Sciences, Johns Hopkins University, Baltimore, United States; [6]Faculty of Health Sciences, University of Macau, Macau, China; [7]Department of Oncology, Johns Hopkins University, Baltimore, United States

*For correspondence:
mparson3@jhmi.edu (MJP);
jmumm3@jhmi.edu (JSM)

[†]Co-corresponding authors, these laboratories contributed equally to this work

Present address: [‡]Department of Ophthalmology, Wilmer Eye Institute, Johns Hopkins University, Baltimore, United States

**Abstract** Whole-organism chemical screening can circumvent bottlenecks that impede drug discovery. However, in vivo screens have not attained throughput capacities possible with in vitro assays. We therefore developed a method enabling in vivo high-throughput screening (HTS) in zebrafish, termed automated reporter quantification in vivo (ARQiv). In this study, ARQiv was combined with robotics to fully actualize whole-organism HTS (ARQiv-HTS). In a primary screen, this platform quantified cell-specific fluorescent reporters in >500,000 transgenic zebrafish larvae to identify FDA-approved (Federal Drug Administration) drugs that increased the number of insulin-producing β cells in the pancreas. 24 drugs were confirmed as inducers of endocrine differentiation and/or stimulators of β-cell proliferation. Further, we discovered novel roles for NF-κB signaling in regulating endocrine differentiation and for serotonergic signaling in selectively stimulating β-cell proliferation. These studies demonstrate the power of ARQiv-HTS for drug discovery and provide unique insights into signaling pathways controlling β-cell mass, potential therapeutic targets for treating diabetes.

## Introduction

Diabetes is associated with reductions in pancreatic β-cell mass, thus, curing diabetic patients will require β-cell replacement therapy. β cells can be replaced by transplantation of pancreatic islets (*Vardanyan et al., 2010*). Alternatively, drugs that induce β-cell differentiation in endogenous pancreatic progenitor cells (*Chong et al., 2006a*) or which stimulate β-cell proliferation (*Lysy et al., 2013*) would provide a significant step forward in the treatment of diabetes by avoiding surgical risks.

Placing animal models at the start, rather than the end, of the drug discovery process has the potential to circumvent high attrition rates that have plagued in vitro high-throughput screening (HTS) over the past two decades. The zebrafish is an ideal vertebrate model system for whole-organism-based drug discovery (*Zon and Peterson, 2005*). As a key example, a chemical derivative of prostaglandin E2 (16,16 dimethyl prostaglandin $E_2$), originally identified for the capacity to induce increased hematopoietic stem cell (HSC) numbers in zebrafish embryos, recently completed Phase I and entered Phase II clinical trials as a means of enhancing engraftment of cord blood transplants in leukemic patients (*Cutler et al., 2013*).

**eLife digest** Type 1 diabetes is caused by the body incorrectly destroying the cells in the pancreas—known as β cells—that produce insulin and so control the amount of sugar found in the bloodstream. Drugs that increase the rate at which new β cells form could therefore help to treat this disease.

High-throughput screening is a technique that uses automated systems to rapidly test the effects of large numbers of drug-like compounds on living cells. Unfortunately, drugs sometimes produce different effects in animals than those they produce in isolated cells or other more simplified screening systems.

Zebrafish are often used in biological studies because the larvae are transparent, making it easier to study what goes on inside them. Wang et al. have now developed a high-throughput screening system that uses genetically engineered zebrafish. The zebrafish contain 'reporter' genes that fluoresce when a gene is activated, and the intensity of the fluorescence can be interpreted to work out the effects of an applied drug.

To search for compounds that cause β cells to grow, Wang et al. created two reporter genes: one that glows yellow when new β cells form, and one that glows red when other pancreatic cells are stimulated. An initial screen tested the effects of over 3000 drugs, most of which have been approved for use in humans. This screen identified and confirmed 24 drugs that trigger the growth of new β cells or other pancreatic cells in zebrafish larvae. Further investigation uncovered new roles for two signaling pathways that had not previously been linked to pancreatic growth. One pathway—the serotonin pathway, which is better known for transmitting signals in the brain—selectively stimulates the growth of new β cells.

The work of Wang et al. therefore presents a number of possible drugs and pathways that could be targeted in the search for a new treatment for type 1 diabetes. Furthermore, this new whole-organism, high-throughput screening system could be used in the future to search for drugs that affect a range of other biological processes.

Conserved cellular and molecular mechanisms are known to govern pancreatic development and β-cell proliferation in zebrafish and mammals (*Zorn and Wells, 2007*; *Kinkel and Prince, 2009*). We therefore hypothesized that identifying pre-existing drugs that promote increased β-cell mass during zebrafish development might provide potential new drug leads and therapeutic targets for treating diabetic patients. In embryonic zebrafish, early endocrine cells exist as a single principal islet in the head of the pancreas. At larval stages, additional endocrine cells are added by differentiation of extra-pancreatic ductal cells and proliferation within the principal islet (*Dong et al., 2007*; *Pisharath et al., 2007*). Around 6 days post-fertilization (dpf), progenitors located in the pancreatic duct start to differentiate to form smaller secondary (2°) islets. At this early stage, 2° islets consist of one or more endocrine cells that form within the tail of the pancreas (*Biemar et al., 2001*; *Wang et al., 2011*; *Ninov et al., 2012*). These easily visualized features of zebrafish pancreatic development can be used to delineate the specific effect(s) of exogenous factors on β-cell biology. For instance, the appearance of 'precocious' 2° islets before 6 dpf is an indication of induced endocrine differentiation (*Rovira et al., 2011*). Conversely, an increase in principal islet cell numbers, in the absence of effects on endocrine differentiation (e.g., 2° islet formation), suggests stimulation of endocrine cell proliferation. We previously visualized precocious 2° islet formation in a manual chemical screen of a library of largely FDA-approved drugs (Johns Hopkins Drug Library; JHDL) to identify six compounds that induced endocrine differentiation (*Rovira et al., 2011*). Here, we developed a reporter-based strategy for identifying compounds that increase β-cell mass at high-throughput rates. By labeling β cells with a fluorescent protein and quantifying changes in fluorescence after exposure to JHDL compounds, many more drugs were identified that induced endocrine differentiation and/or stimulated proliferation of β cells.

Most whole-organism drug discovery efforts to date have relied on manual assays or high-content screening (HCS). These approaches attain only mid-throughput rates, thus, in vivo drug screens have typically been limited to small sample sizes and screening compounds at a single concentration (*Mathias et al., 2012*). Ideally, false-call rates could be minimized by using 'statistical power' to

establish appropriate sample sizes (*Ellis, 2010*; *Grissom and Kim, 2011*; *Halsey et al., 2015*) and by testing compounds at multiple concentrations, a strategy called 'quantitative HTS' (qHTS; [*Inglese et al., 2006*]). However, due to increased volume demands, applying such strategies to whole-organism drug discovery requires methods for evaluating compounds at HTS rates in vivo. Toward that end, we previously adapted existing HTS instrumentation, specifically a microplate reader, to the task of quantifying fluorescent reporters in living zebrafish, termed automated reporter quantification in vivo (ARQiv; (*Walker et al., 2012*). ARQiv provides purely quantitative data whereas HCS typically produces images, thus, more complex data. However, offsetting any comparative reduction in data complexity, ARQiv significantly increases throughput capacity. Indeed, ARQiv assays can be performed at a pace equivalent to in vitro HTS; an upper limit of 200,000 organisms per day, per plate reader (*Walker et al., 2012*). Accordingly, ARQiv enables optimal HTS practices, such as qHTS, to be applied to whole-organism drug discovery.

Here, we have combined ARQiv with a custom-designed robotics system to enable the first truly high-throughput whole-organism drug screen in a vertebrate model (ARQiv-HTS). We analyzed a zebrafish transgenic line in which β cells are labeled with (Yellow Fluorescent Protein) YFP and neighboring delta (δ) cells are labeled with RFP (*Walker et al., 2012*). The goal of the primary screen was to identify drugs that increased β-cell reporter activity relative to vehicle only controls, thus, compounds that potentially increased β-cell mass. Secondary confirmation screens were designed to determine whether potential hit drugs induced endocrine differentiation (precocious secondary islet formation) or stimulated β-cell proliferation (increased β-cell numbers in the absence of effects on differentiation). Our results revealed: (1) ARQiv can be applied at HTS rates. Over 500,000 transgenic larvae were evaluated in the primary screen and can detect small differences in the number of fluorescently labeled cells; (2) qHTS can be effectively applied to whole-organism drug discovery. All JHDL compounds were tested at six different concentrations and a sample number of 16 per condition; (3) new purposes for FDA-approved drugs in increasing β-cell mass. We validated 11 drugs that induced endocrine differentiation and 15 drugs that stimulated β-cell proliferation (two compounds had activity in both assays); and (4) novel roles for NF-κB signaling in regulating pancreatic progenitor differentiation and for serotonergic signaling in selectively stimulating β-cell proliferation. Due to the near limitless number of reporter-based assays applicable to ARQiv-HTS—that is, anything involving a change in reporter intensity—we anticipate this approach will become a useful platform for whole-organism drug discovery and development.

## Results

### Establishing an HTS-compatible ARQiv assay

Our first goal was to develop a HTS-compatible reporter-based assay for identifying compounds that increase pancreatic β-cell mass in vivo. Toward that end, we established a dual-reporter transgenic line, *Tg(ins:PhiYFP-2a-nsfB, sst2:tagRFP)lmc01* (β/δ-reporter) in which the *insulin* (*ins*) promoter drives expression of a yellow fluorescent protein (PhiYFP) in β cells, and the *somatostatin 2* (*sst2*) promoter drives a red fluorescent protein (TagRFP) in adjacent δ cells (*Figure 1A,B*; (*Walker et al., 2012*). We reasoned that the β/δ-reporter line would allow us to detect compounds affecting endocrine differentiation and/or proliferation of β cells or their progenitors since both would cause an increase in YFP reporter signal (*Figure 1C*). Expressing RFP in δ cells secondarily could facilitate identification of compounds that selectively increased β-cell mass (>YFP only) vs expansion of endocrine tissue in general (>YFP and >RFP). We tested whether a chemical inhibitor of γ-secretase (DAPT), an enzyme necessary for Notch signaling, could serve as a positive control. Prior studies had shown that inhibition of Notch signaling promoted precocious 2° islet formation and thereby increased *insulin* reporter activity (*Parsons et al., 2009*). We therefore adapted a protocol used to manually screen for precocious 2° islet formation at 5 dpf (*Rovira et al., 2011*) to the task of detecting increased β-cell numbers (>YFP fluorescence) via ARQiv.

To determine an optimal dosage, DAPT was titrated across a twofold dilution series (from 200 μM to 6.25 μM) and used to treat β/δ-reporter larvae for 2, 3, and 4 days starting at 3 dpf. Reporter signals induced by DAPT treatment were compared to vehicle only negative controls (0.1% DMSO). This analysis determined that a 4-day exposure (3–7 dpf; *Figure 1D*) achieved reporter signal levels necessary for HTS. The data also validated the utility of DAPT as a positive control for inducing increased YFP signal (maximal DAPT/DMSO ratio of >5.5) and to a lesser extent for RFP (maximal

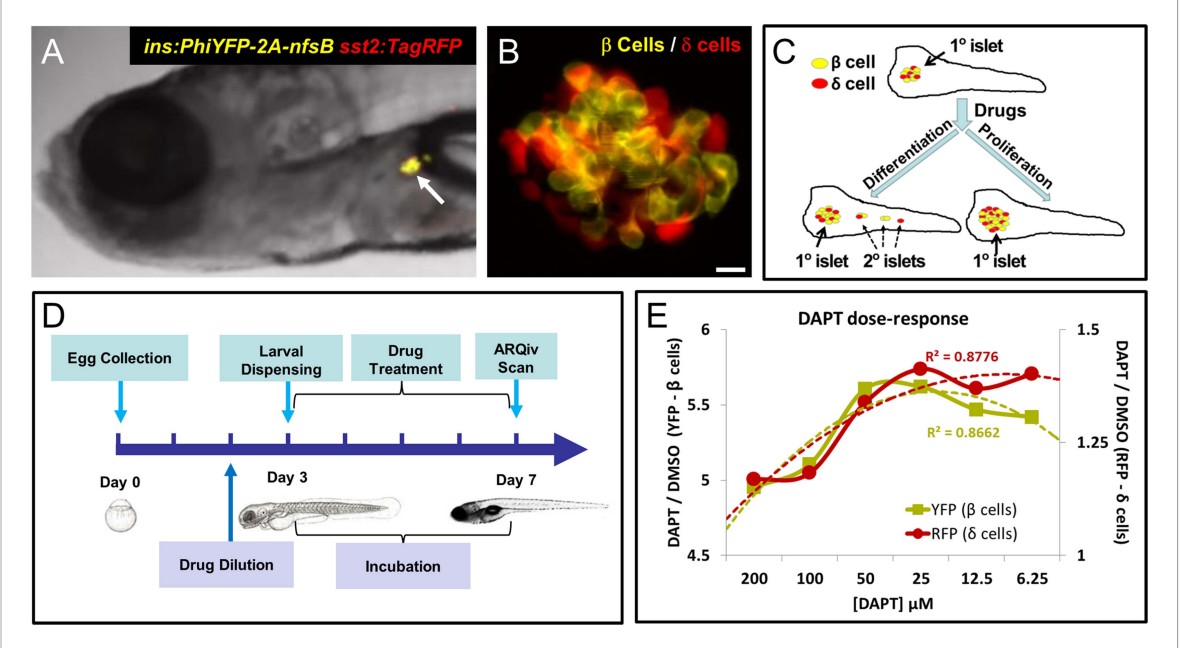

**Figure 1**. Screening resources, design, and controls. (**A**) Transgenic line used for the primary screen, *Tg(ins:PhiYFP-2a-nsfB, sst2:tagRFP)lmc01* (β/δ reporter; *Walker et al., 2012*), the *insulin* promoter drives YFP-expression in β cells (yellow), the *somatostatin 2* promoter drives RFP expression in neighboring δ cells (red). Photomicrograph of the anterior region of a 7 dpf larva shows YFP and RFP labeling of the principal islet (arrow). (**B**) Confocal z-projection of the principal islet in a β/δ-reporter fish (scale bar: 10 µM), YFP labeling β cells (yellow) and RFP labeling δ cells (red)—note, apparent 'orange' co-labeling is an artifact of z-projection in 2D format. (**C**) Illustration of two potential mechanisms by which drug exposures could lead to increased β-cell mass: (1) enhanced endocrine differentiation, indicated by secondary (2°) islet formation (left path) and (2) increased β-cell proliferation, indicated by supernumerary β cell numbers in the principal islet (right path) in the absence of effects on endocrine differentiation—that is, no effect on 2° islet formation. (**D**) Schematic of the ARQiv-HTS screening process: Day 0, mass breeding produced 5000–10,000 eggs per day; Day 2 (evening), JHDL compounds were serially diluted into drug plates; Day 3, the COPAS-XL (Union Biometrica) was used to dispense individual 3 dpf larvae into single wells of drug plates, and plates were then maintained under standard conditions for 4 days; Day 7, larvae were anesthetized and reporters quantified by automated reporter quantification in vivo (ARQiv). (**E**) β/δ-reporter larvae were exposed to 0.1% DMSO (negative control) or the γ-secretase/Notch inhibitor DAPT (positive control) at six different concentrations from 3 to 7 dpf. ARQiv was then used to measure fluorescent signals from β cells (yellow line, left y-axis) and δ cells (red line, right y-axis). The DAPT to DMSO ratio (DAPT/DMSO) was used to indicate signal strength for each fluorophore independently, as per the primary screen. The β-cell data show a non-monotonic dose response (yellow dashed line, polynomial curve fit), with maximal signal observed at 25–50 µM DAPT. The δ-cell data show a similar trend (red dashed line, polynomial curve fit), but with approximately fourfold lower signal strength due to higher autofluorescent background in the RFP emission range.

The following figure supplement is available for figure 1:

**Figure supplement 1**. ARQiv-HTS system.

DAPT/DMSO ratio of >1.25, see *Figure 1E*). Dose-response curves show concentration-dependent effects for both cell types, with maximal responses at 25–50 µM.

To assess assay quality, establish appropriate sample sizes, and set 'hit' call criteria, we used statistical methods developed for HTS that account for increased signal variability attending in vivo assays (see 'Materials and methods', and [*White et al., 2015*]). To generate large data sets for this analysis, 192 individual positive (DAPT) and negative (DMSO) control assays were performed. Strictly standardized mean difference (SSMD) calculations were used to determine assay quality, set a hit call cut-off, and as a means of comparing effect size across compounds (*Zhang, 2011*). This analysis determined that our assay was of high enough quality to pursue HTS (robust SSMD* score of 1.67). The sample size calculation (*Ellis, 2010*; *Grissom and Kim, 2011*), using power and significance values minimizing false-call rates (99.9% and p = 0.001, respectively), determined that a sample number of 14 was sufficient to detect a 50% effect size (i.e., half as potent as the DAPT positive control). However, to account for occasional automation errors, and in keeping with 96-well plate layouts, we elected to

screen 16 larvae per compound concentration. Due to greater background autofluorescence in the RFP emission range, a sample size of 16 was predicted to be insufficient for detecting a 50% effect size on δ cells. Thus, we limited the use of RFP data to a simple comparison between YFP and RFP dose-responses, rather than as a ratiometric standard. Bootstrapping (random sampling with replacement) of the positive and negative control data sets at a sample size of 16 resulted in a predicted SSMD score of 1.3 for an effect size of 50% relative to the positive control. Accordingly, we set the SSMD 'hit' selection cut-off at ≥1.3.

## Primary screen: ARQiv assay

After defining the sample size and hit criterion, we initiated a full-scale screen of the JHDL (*Chong et al., 2006b, 2006c*) using the ARQiv-HTS system (*Figure 1—figure supplement 1A,B*). The JHDL is a collection of 3348 compounds, comprised largely of drugs approved for use in humans (*Shim and Liu, 2014*). Screening the JHDL served three purposes: (1) tested the value of whole-organism qHTS by screening the same library as our prior manual screening effort (*Rovira et al., 2011*), (2) provided an enriched number of biologically active compounds with defined mechanisms of action, and (3) facilitated the identification of existing drugs as potential new treatments for diabetes. Moreover, drug repurposing has the potential to fast track delivery of new therapeutics to the clinic (*Shim and Liu, 2014*).

Custom-designed mass breeding units were used to maximize egg production (*White et al., 2015*). The number of viable eggs on day 1 established the number of drugs to be tested per session. The evening of day 2, robotic plate and liquid handling systems (Hudson Robotics) were used to titrate all JHDL compounds across a twofold dilution series from 4 µM to 125 nM in 0.1% DMSO, thus, testing a total of six different concentrations (*Figure 1D*) per qHTS principles (*Inglese et al., 2006*). At a sample size of 16 per condition, this equated to each drug being arrayed across an entire 96-well plate. DAPT and DMSO control plates bracketed each subset of 10 drug plates (*Figure 1—figure supplement 1C*). On day 3, the COPAS-XL system (Complex Object Parametric Analyzer and Sorter, Union Biometrica) was used to automate dispensing of individual 3 dpf β/δ-reporter larvae into single wells of 96-well plates containing pre-diluted drug solutions; all plates were then incubated under standard conditions. After a 4-day treatment regimen, 7 dpf larvae were anesthetized and fluorescent reporter levels were quantified by ARQiv (*Figure 1D*). We developed an R script for processing and plotting ARQiv data in near real-time to flag plates containing potential hit compounds. This was done to facilitate immediate visual follow-up of larvae in potential hit plates using standard microscopy to eliminate false positives, such as increased autofluorescence due to toxicity, and as an initial assessment of effects on 2° islet formation. Three graphical outputs were plotted: (1) standard box plots, Drug over DMSO (Drug/DMSO) signal ratios to reveal dose-responses and variability, (2) SSMD hit scores, as a means of comparing effect size and to flag compounds of interest (those in which at least one concentration achieved an SSMD ≥1.3), and (3) heat maps, to guide initial visual follow-ups to assess 2° islet formation (*Figure 2A–C*).

After screening more than 500,000 β/δ-reporter larvae, 225 compounds (6.7%) produced an SSMD ≥1.3 and were designated as 'hit calls' (*Figure 2D*). Corresponding plates underwent an initial visual assessment. 29 hit call compounds proved to be autofluorescent, another 19 negatively impacted fish viability and/or morphology. These 48 compounds were designated as false positives and eliminated from further evaluation (*Figure 2D*). The remaining 177 hit call plates were further examined for evidence of enhanced 2° islet formation (*Parsons et al., 2009*; *Rovira et al., 2011*). Increased 2° islet formation was observed in 23 plates (*Figure 2D*, 'Hit I' subset; *Figure 2—figure supplement 1*, *Supplementary file 1*). These 23 Hit I compounds were deemed most relevant for secondary validation assays involving a more direct test of endocrine differentiation effects, precocious 2° islet induction (*Parsons et al., 2009*; *Rovira et al., 2011*). The other 154 hit call plates displayed no preliminary evidence of enhanced 2° islet formation. To account for other mechanisms that could result in elevated insulin reporter activity, we examined a second group of 23 strongly implicated drugs (SSMD values ≥1.75, *Figure 2D*, 'Hit II' subset) for evidence of increased β-cell numbers in the absence of differentiation effects (*Figure 2E*; *Supplementary file 1*). A residual 131 compounds await further evaluation (*Supplementary file 2*). The majority of the 46 Hit I and Hit II compounds underwent a series of 'validation assays' to confirm effects on endocrine differentiation and/or β-cell proliferation. In keeping with common HTS practices, secondary assays were performed with complementary toolsets rather than the β/δ-reporter line used in the primary screen in order to independently confirm the findings of the primary screen.

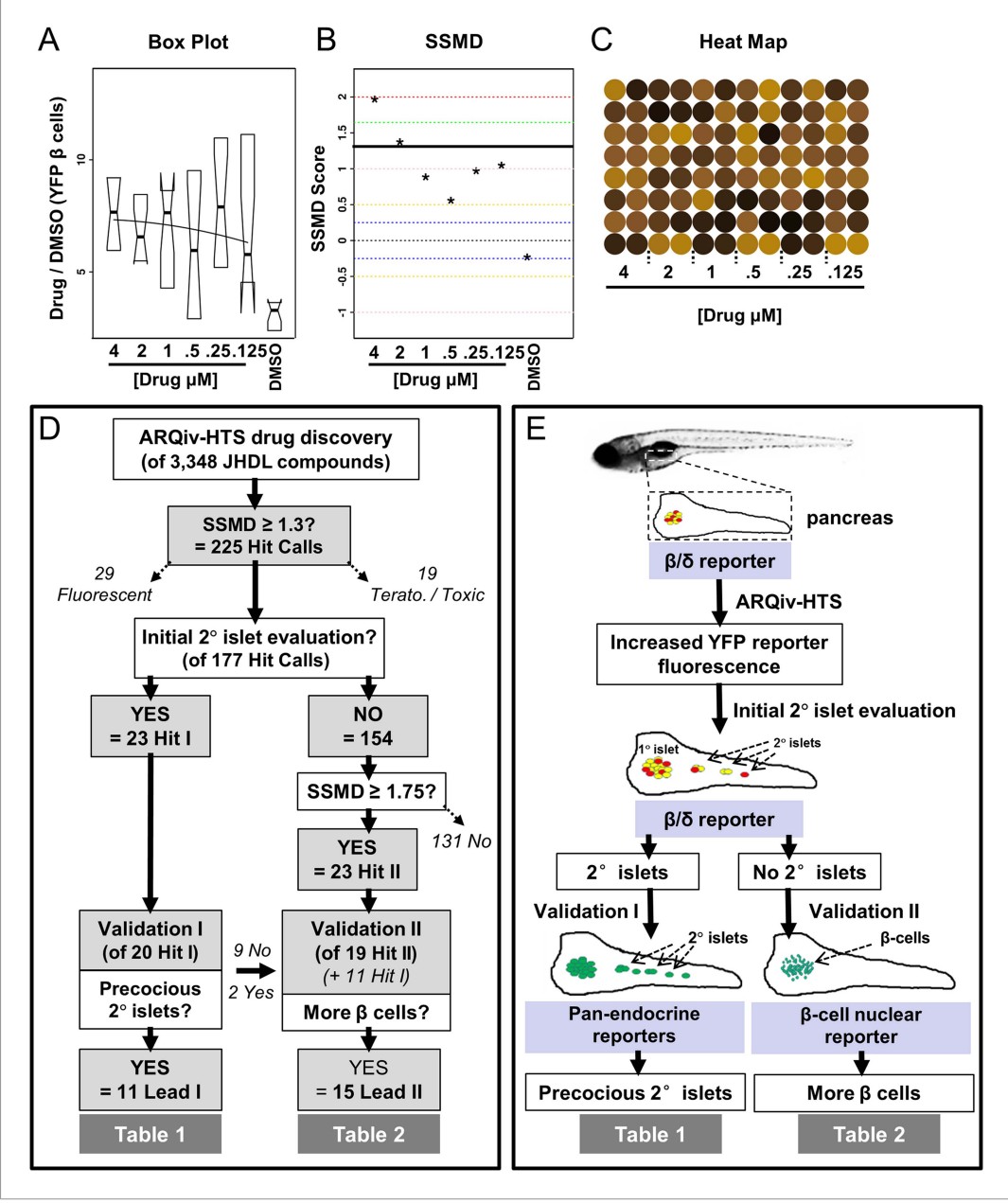

**Figure 2**. ARQiv data and screen flow chart. (**A–C**) Example of MATLAB/R-generated real-time data plots provided for each drug plate; note, data for YFP are shown; however, plots were provided for both fluorophores. (**A**) Boxplots of Drug to DMSO signal ratio (Drug/DMSO) provided dose–response and variance data. (**B**) Strictly standardized mean difference (SSMD) scores were used to rank compounds according to relative strength; black line shows the 1.3 cut-off used to implicate compounds of interest (i.e., 'hit calls'). (**C**) Heat maps facilitated same day visual evaluation of each hit call plate. (**D**) Screening process—drug discovery results: the numbers of compounds tested, implicated (Hits I and II), and validated (Leads I and II) are listed at each stage. In addition, hit calls that were eliminated from further analysis due to being either fluorescent (29 compounds) or toxic (19 compounds), and others which remain to be further evaluated (131 compounds), are indicated by diagonal dashed arrows. (**E**) Screening process—assays utilized: schematic showing primary and secondary screening processes. In keeping with the high-throughput screening (HTS) practice of confirming implicated compounds in 'orthogonal' assays, different transgenic reporter lines were used for the following (progressing from the top): (1) the primary screen and initial 2° islet evaluation (β/δ-reporter), (2) validating effects on endocrine differentiation (pan-endocrine GFP reporters), and (3) validating of effects on β-cell proliferation (β-cell nuclear reporters).

*Figure 2. continued on next page*

*Figure 2. Continued*

The following figure supplement is available for figure 2:

**Figure supplement 1**. Observation of 2° islet formation in live β/δ-reporter larvae after drug treatment.

## Validation assay I: endocrine differentiation

In preliminary visual evaluations to assess endocrine differentiation effects in larvae within hit call plates, considerable variability was noted. Typically, additional 2° islets were observed only in a subset of treated larvae among a given hit call condition (*Figure 2—figure supplement 1C*). This is likely due to the β/δ-reporter being less than ideal for detecting endocrine differentiation effects; reporters are linked to late-stage differentiation of β and δ cells, which are only just beginning to appear in 2° islets at 7 dpf (*Parsons et al., 2009*). The requirement of a 4-day chemical exposure to observe expression differences in the β/δ-reporter line via ARQiv reflects this issue. Conversely, we have shown that transgenic lines labeling early endocrine progenitors are useful for identifying compounds that induce endocrine differentiation as early as 5 dpf (*Rovira et al., 2011*). Therefore, in keeping with the practice of using 'orthogonal' assays to confirm the activity of compounds implicated in primary screens (*Thorne et al., 2010*), we tested Hit I compounds in transgenic backgrounds better suited to visualizing 2° islets. In particular, we used the pan-endocrine reporter line, *Tg(neurod:EGFP)nl1* (*Obholzer et al., 2008*; Dalgin and Ward, 2011), to confirm the efficacy of 'Hit I' drugs for inducing early endocrine differentiation. In this line, GFP is expressed in nascent endocrine cells, permitting the detection of 'precocious' 2° islet formation at 5 dpf after 2-day drug exposures, akin to our previous manual screen (*Rovira et al., 2011*).

A subset of Hit I compounds (20 of 23) was tested accordingly. An alternative Notch pathway inhibitor, RO4929097 (5 μM, Selleck Chemicals; *Luistro et al., 2009*), was used as the positive control. Our prior studies had shown that RO4929097 functions equivalently to DAPT for this assay (*Huang et al., 2014*). Transgenic larvae were treated from 3 to 5 dpf with compounds across an expanded concentration range (0.5–25 μM) to account for differences with the primary assay (e.g., transgenic line used, timing and duration of compound treatment) and/or differences between compound lots. Following treatments, larvae were fixed at 5 dpf and processed for imaging by confocal microscopy. As GFP expression is widespread throughout the endoderm in the *neurod:EGFP* line, pancreata of treated fish were micro-dissected. High-resolution imaging afforded an increased sensitivity in scoring the induction of 2° islets (*Figure 3A,B*). Of the 20 Hit I drugs tested, 11 were validated as inducers of endocrine differentiation (55%; *Figure 3C*, *Figure 3—figure supplement 1*). The confirmed hits were reclassified as the 'Lead I' drugs (*Figure 2D*; *Table 1*). Equivalent tests using a second pan-endocrine transgenic line, *Tg(pax6b:GFP)ulg515* (*Delporte et al., 2008*), confirmed the same 11 drugs as leads (*Figure 3—figure supplement 2*).

## Validation assay II: quantification of β-cell number

Of the original ARQiv Calls, 154 compounds showed no preliminary evidence of enhanced 2° islet formation. However, many of these drugs had high SSMD scores suggesting substantial biological significance. We hypothesized that increased β-cell mass in the principal islet would also have been reported as increased *insulin* reporter activity (YFP) during the primary screen. Furthermore, any increase in β-cell mass in the absence of effects on endocrine differentiation would suggest a capacity to induce β-cell proliferation (*Figures 1C, 2D,E*). Discovery of drugs promoting β-cell proliferation would have obvious implications for treating diabetic conditions associated with β-cell paucity.

To prioritize which drugs to screen for increases in β-cell number within the principal islet, we chose a threshold SSMD score of 1.75, denoted as the Hit II subset (*Figure 2D*). Of 23 drugs that met this criterion, we were able to perform validation assays on 19. We also included the 9 Hit I drugs that failed to induce 2° islet formation, and two validated Hit I compounds with an SSMD >1.75; thus, a total of 30 compounds (*Figure 2D*). In β/δ-reporter transgenic fish, YFP is expressed in β-cell cytoplasm, making it difficult to count cell numbers accurately, (*Figure 1B*). To facilitate detailed quantification of β-cell numbers, we turned to a transgenic line in which GFP is expressed in β-cell nuclei, *Tg(ins:hmgb1-EGFP)jh10* (*Wang et al., 2011*). All compounds were tested as per the treatment

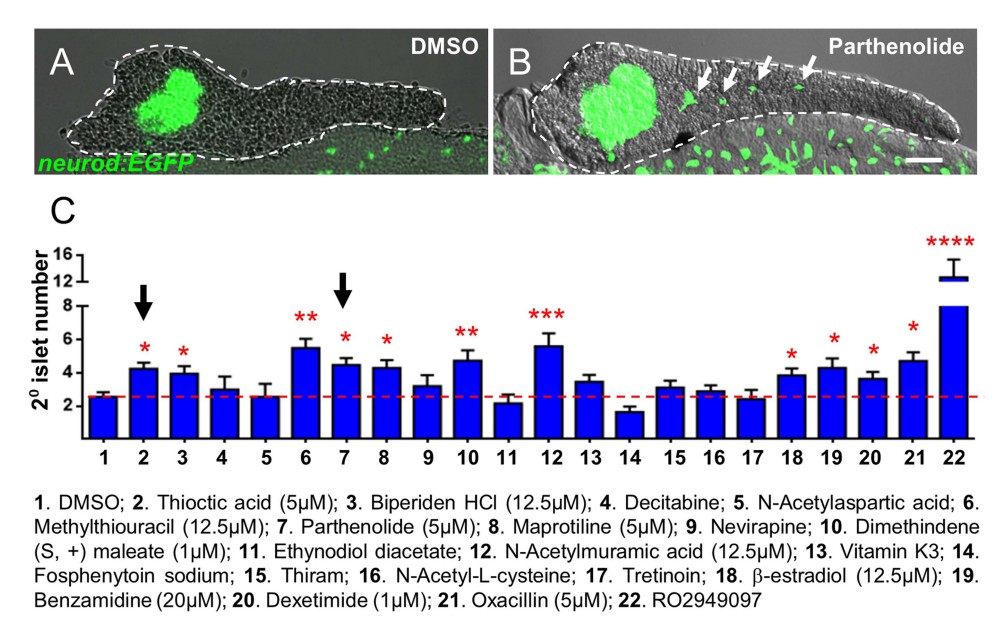

**Figure 3**. Validation of endocrine differentiation induction: precocious 2° islet assay. (**A**, **B**) Representative confocal images—brightfield and fluorescence images merged—of dissected pancreata (dashed lines) from *neurod:EGFP* transgenic larvae treated from 3 to 5 dpf with 0.1% DMSO (**A**) or a Hit I drug (**B**, example shown is parthenolide). Early endocrine cells are labeled with GFP (green) allowing precocious formation of 2° islets (white arrows) to be visualized following drug exposures. (**C**) The number of precocious 2° islets was quantified following treatment with the indicated Hit I compounds from 3 to 5 dpf. Results obtained with the optimal concentration were plotted relative to negative (0.1% DMSO) and positive controls (RO2949097, 5 μM). Of 20 Hit I compounds tested, 11 were confirmed as Lead I drugs for inducing endocrine differentiation (optimal concentrations for validated leads are shown in parentheses). Arrows indicate drugs that inhibit NF-κB signaling. Scale bar, 25 μm. Error bars, standard error. All p-*values* were calculated using Dunnett's test. *p < 0.05, **p < 0.01, ***p < 0.001, ****p < 0.0001. n = 5–10 larvae per condition, experiment was repeated 3 times per compound.

The following figure supplements are available for figure 3:

**Figure supplement 1**. Validation of endocrine differentiation induction: precocious 2° islet assay (*neurod* reporter).

**Figure supplement 2**. Validation of endocrine differentiation induction: precocious 2° islet assay (*pax6b* reporter).

regimen established for Validation assay I. The data showed that 15 compounds (50%) caused a significant increase in β-cell number at the optimal tested concentration (*Figure 4A*). Hits confirmed for the ability to stimulate increased β-cell numbers in the absence of effects on differentiation were reclassified as 'Lead II' drugs (*Figure 4A*; *Table 2*).

## Selective stimulation of β-cell proliferation

Of the leads that increased β-cell number, paroxetine was particularly intriguing as comparisons between β-cell and δ-cell reporter activity suggested that this drug might selectively increase β-cell numbers without affecting δ-cells, that is, potentially acting in a cell type-specific manner (*Figure 4B,C*). To verify that the actions of paroxetine were specific to β cells, we treated double transgenic (*ins: hmgb1-EGFP; β/δ-reporter)* larvae with paroxetine at the optimal concentration (1 μM) from 3 to 5 dpf and quantified β and δ cells using confocal microscopy. DMSO treated larvae had an average of 29.4 ± 1.1 β cells and 24.1 ± 1.6 δ cells. As expected, RO4929097 treatment caused a significant increase in both endocrine cell types examined (35.8 ± 1.3 β cells and 29.7 ± 1.5 δ cells; *Figure 4D–H*), consistent with DAPT treatment in the ARQiv assay (*Figure 1E*) and likely due to induced differentiation of progenitors contributing to the principal islet. Conversely, paroxetine significantly increased β-cell

**Table 1.** Lead I drugs: inducers of endocrine differentiation

| | Drug name | >2° islet # | ARQiv (µM)* | >2° islet (µM)* |
|---|---|---|---|---|
| 1 | N-Acetylmuramic acid | +++ | 0.5 | 12.5 |
| 2 | Methylthiouracil | ++ | 0.25 | 12.5 |
| 3 | Dimethindene (S, +) maleate | ++ | 1 | 1 |
| 4 | Thioctic acid | + | 0.5 | 5 |
| 5 | Biperiden HCl | + | 1 | 12.5 |
| 6 | Parthenolide | + | 4 | 5 |
| 7 | Maprotiline | + | 0.5 | 5 |
| 8 | Estradiol diacetate → Beta-estradiol | + | 1 | 12.5 |
| 9 | Oxacillin | + | 1 | 5 |
| 10 | Benzamidine | + | 0.125 | 20 |
| 11 | Dexetimide | + | 1 | 1 |
| 12 | Decitabine | – | 0.5 | n/a |
| 13 | N-Acetylaspartic acid | – | 4 | n/a |
| 14 | Nevirapine | – | 0.25 | n/a |
| 15 | Ethynodiol diacetate | – | 0.25 | n/a |
| 16 | Vitamin K3 | – | 0.25 | n/a |
| 17 | Fosphenytoin sodium | – | 0.25 | n/a |
| 18 | Thiram | – | 0.5 | n/a |
| 19 | BOC-S-acetaminomethyl-L-cysteine→ N-Acetyl-L-cysteine | – | 0.5 | n/a |
| 20 | Tretinoin | – | 0.25 | n/a |
| 21 | Iodine | nd | 0.25 | nd |
| 22 | Bayberry wax | nd | 0.25 | nd |
| 23 | 1,5-Bis (succinimidooxycarbonyloxy) pentane | nd | 0.5 | nd |

The 23 Hit I drugs are listed. 20 were tested for induction of endocrine differentiation, that is, precocious 2° islet formation. Compounds are ordered according to the results of the validation screen, 11 drugs were confirmed as leads (++ = p < 0.01'; + = p < 0.05), 9 failed (–). *optimal response concentration for the ARQiv and validation screens. n/a: not applicable; nd: not determined, n = 5–10 larvae per condition, experiment repeated 3 times.

numbers (37.1 ± 1.4, p < 0.01) but had no effect on δ cells (23.0 ± 1.1, p = 0.57) (*Figure 4D–H*). This result suggests that it is possible to increase β-cell mass without incurring concomitant increases in other endocrine compartments, an important finding with respect to the development of targeted therapies.

## Mechanism of action studies

One of the central advantages of screening clinically approved drugs is that molecular mechanisms of action are typically well characterized. Thus, having validated several compounds for the capacity to induce endocrine differentiation (*Table 1*) and/or β-cell proliferation (*Table 2*), we next sought to investigate whether shared mechanisms of action were implicated between drugs eliciting the same effect on pancreatic biology, that is, among compounds within the Lead I or Lead II sets.

## NF-κB signaling regulates endocrine differentiation

Two of the 11 drugs in the 'Lead I' set, thioctic acid and parthenolide, are known inhibitors of the NF-κB signaling pathway (*Ying et al., 2010*; *Ghantous et al., 2013*). This inspired us to ask whether these drugs enhance endocrine differentiation by modulating NF-κB signaling. In quiescent cells, the NF-κB complex is sequestered in the cytoplasm and associates with inhibitory IκB proteins (*Schmitz et al., 2004*).

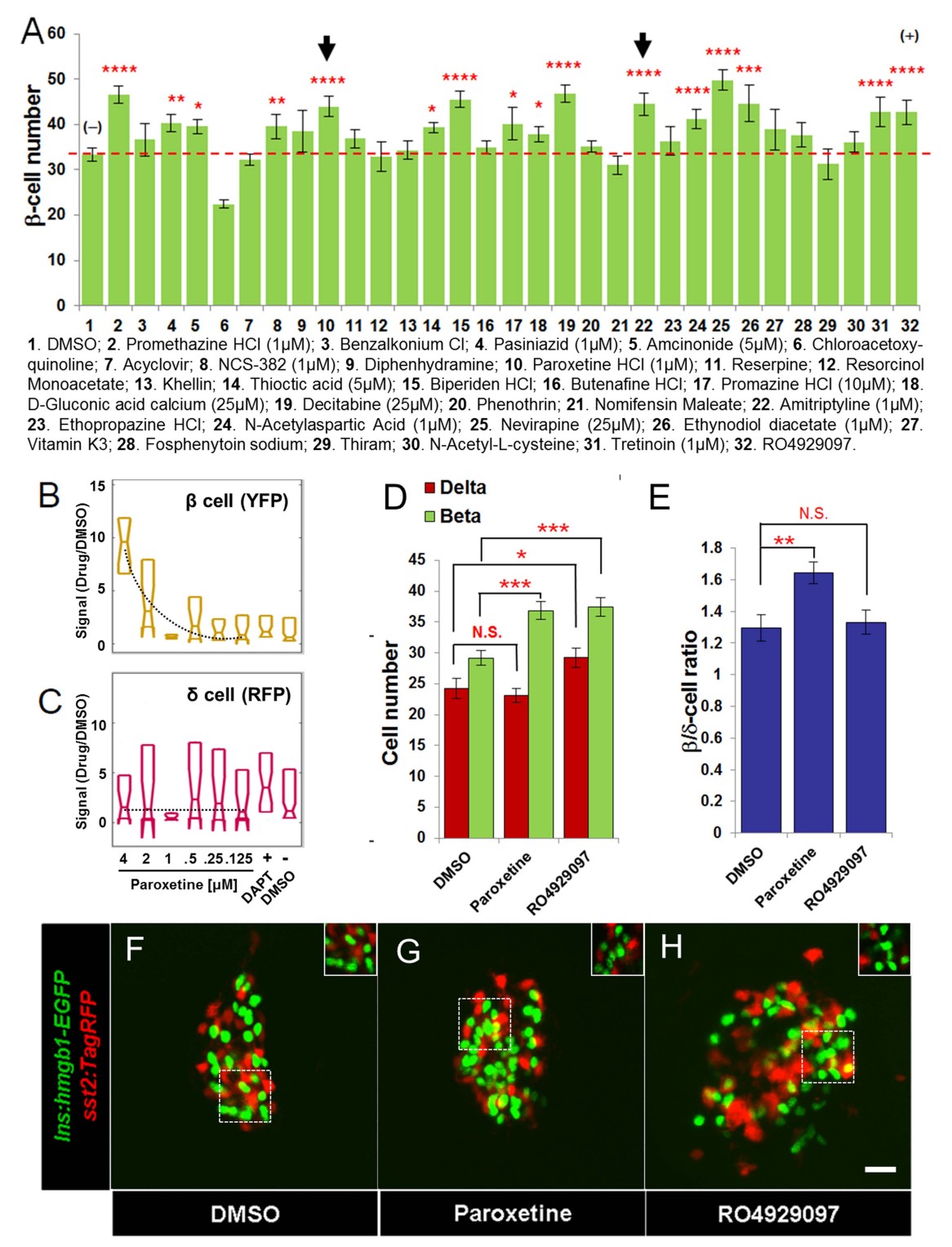

1. DMSO; 2. Promethazine HCl (1μM); 3. Benzalkonium Cl; 4. Pasiniazid (1μM); 5. Amcinonide (5μM); 6. Chloroacetoxy-quinoline; 7. Acyclovir; 8. NCS-382 (1μM); 9. Diphenhydramine; 10. Paroxetine HCl (1μM); 11. Reserpine; 12. Resorcinol Monoacetate; 13. Khellin; 14. Thioctic acid (5μM); 15. Biperiden HCl; 16. Butenafine HCl; 17. Promazine HCl (10μM); 18. D-Gluconic acid calcium (25μM); 19. Decitabine (25μM); 20. Phenothrin; 21. Nomifensin Maleate; 22. Amitriptyline (1μM); 23. Ethopropazine HCl; 24. N-Acetlaspartic Acid (1μM); 25. Nevirapine (25μM); 26. Ethynodiol diacetate (1μM); 27. Vitamin K3; 28. Fosphenytoin sodium; 29. Thiram; 30. N-Acetyl-L-cysteine; 31. Tretinoin (1μM); 32. RO4929097.

**Figure 4.** Validation of increased β-cell proliferation: cell counts. (**A**) Quantification of β-cell numbers following incubation of *ins:hmgb1-EGFP* transgenic larvae from 3 to 5 dpf in one of 30 Hit compounds, 0.1% DMSO, or the Notch inhibitor RO4929097 (5 μM). 15 compounds were confirmed as Lead II drugs for increasing β-cell numbers. Arrows indicate drugs that enhance serotonin signaling. (**B**, **C**) ARQiv screen data for paroxetine: box plots of β cells (**B**) and δ cells (**C**) suggest a β cell-specific effect—that is, a dose–response in YFP but not RFP signal (dashed line, single polynomial curve fit). (**D**) Numbers of δ cells (red bars) and β cells (green bars) were quantified following treatment with paroxetine, 0.1% DMSO, or RO4929097. Increased β-cell numbers were
*Figure 4. continued on next page*

*Figure 4. Continued*

seen following paroxetine and RO4929097 treatments. However, only RO4929097 increased both β and δ cells. (**E**) Ratio of the number of β cells to δ cells, which confirms that the number of β cells increases following paroxetine treatment relative to δ cells, suggesting cell-type selective effects. Error bars, standard error. n = 5–10 larvae per condition, experiment was repeated 2–3 times per compound. (**F–H**) Representative z-projection confocal images of the principal islets in dissected pancreata (post-paraformaldehyde fixation) from*Tg(ins:hmgb1-EGFP; β/δ-reporter*) triple transgenic lines treated with DMSO (**F**), paroxetine (**G**), or RO4929097 (**H**). Shown are EGFP+ β-cell nuclei (green) and TagRFP+ δ cells (red); note, PhiYFP in the β/δ-reporter line does not withstand fixation, allowing 'clean' labeling of β-cell nuclei with EGFP. In addition, apparent overlap between the β-cell and δ-cell markers (i.e., occasional 'yellow' cells) is an artifact of z-projection images shown in 2D format. For clarity, the inset panels show a single z-slice image of partial islet showing no co-localization of cell type specific reporters. All p-*values* were calculated using Dunnett's test. *p < 0.05, **p < 0.01, ***p < 0.001, ****p < 0.0001. N.S., non-significant. Scale bar, 10 µm.

Activation of NF-κB signaling leads to kinase-dependent phosphorylation and degradation of IκB, allowing NF-κB to translocate to the nucleus and regulate target-gene transcription (*Schmitz et al., 2004*). Despite having different molecular targets, thioctic acid and parthenolide both block NF-κB nuclear translocation (*Ying et al., 2010*; *Ghantous et al., 2013*). To further validate the NF-κB pathway as a target for stimulating endocrine differentiation, we used two other compounds, not present in our chemical library, but known to inhibit NF-κB signaling at different steps in the pathway: NF-κB inhibitor II (NFκBi-II) blocks the target transcription without affecting IκB degradation (*Shin et al., 2004*); and NF-κB I inhibitor III (NFκBi-III) inhibits cytokine-stimulated NF-κB activation (*Lee et al., 2005*). *Neurod:GFP* larvae were used to determine effects on 2° islet appearance following incubation (3–5 dpf) with either NFκBi-II (1 µM) or NFκBi-III (10 µM). Initial working concentrations of these compounds were based on previous studies (*Shin et al., 2004*; *Lee et al., 2005*). Dosages were then decreased until concentrations that did not induce morphological defects were defined. Treated larvae showed significant increases in the 2° islet number with both NFκB inhibitors (*Figure 5A–D*). From this work, it is clear that targeting multiple steps in the NF-κB signaling pathway results in precocious 2° islet formation, and therefore, β-cell neogenesis. Using a previously characterized NF-κB reporter transgenic line, *Tg(6xNFκB: EGFP)nc1* (*Kanther et al., 2011*), we confirmed that 2-day treatments (3–5 dpf) with thioctic acid, parthenolide, as well as NF-κB inhibitors II and III dramatically reduced NF-κB reporter activity in the pancreas and globally (*Figure 5—figure supplement 1*). We next sought to confirm that NF-κB inhibition could induce endocrine differentiation. Larvae from the pan-endocrine reporter line, *neurod:EGFP* (*Figure 5A–C*), showed a significant increase in 2° islet number when treated with either inhibitor. This result clearly demonstrates that targeting the NF-κB signaling pathway results in precocious 2° islet formation, thus, β-cell neogenesis through induction of endocrine differentiation.

Having established that NF-κB is involved in the regulation of endocrine differentiation, we sought to identify which pancreatic cell types are actively undergoing NF-κB signaling during development. To do so, pancreata from double transgenic fish carrying NF-κB and Notch pathway reporters (*6xNFκB:EGFP; tp1:hmgb1-mCherry*) (*Parsons et al., 2009*; *Kanther et al., 2011*) were imaged using confocal microscopy. We found that the NF-κB reporter signal (GFP) overlapped with the Notch-pathway dependent signal (mCherry) at 5 dpf (*Figure 5D,D'*). This indicates that NF-κB signaling is active in Notch-responsive progenitors that line the pancreatic duct, consistent with a novel role for the NF-κB pathway in endocrine differentiation. Of note, NF-κB inhibition did not appear to reduce Notch-reporter expression (*Figure 5—figure supplement 1A–E*), a result requiring further characterization.

## Serotonergic signaling stimulates β-cell proliferation

A potential shared mechanism of action among Lead II compounds was revealed by the fact that paroxetine and amitriptyline (*Figure 4A*, compounds 10 and 22, arrows; *Table 2*) are both predicted to increase serotonergic signaling (*Dechant and Clissold, 1991*; *Boyer and Feighner, 1992*) (*Sangdee and Franz, 1979*). Clinically, both drugs are used as antidepressants. Paroxetine is a selective serotonin reuptake inhibitor (SSRI), thereby increasing extracellular serotonin concentration. Amitriptyline inhibits reuptake of both norepinephrine and serotonin.

We hypothesized that these drugs regulated β cells by mediating serotonergic signaling. To test this hypothesis, we evaluated fluoxetine, another SSRI, and serotonin itself. β cells were quantified

**Table 2**. Lead II drugs: increased β-cell number

| | Drug name | > β-cell # | ARQiv (μM)* | > β-cell (μM)* |
|---|---|---|---|---|
| 1 | Promethazine HCl | ++++ | 0.125 | 1 |
| 2 | Paroxetine HCl | ++++ | 4 | 1 |
| 3 | Biperiden HCl | ++++ | 1 | 12.5 |
| 4 | Decitabine | ++++ | 0.5 | 25 |
| 5 | Amitriptyline | ++++ | 2 | 1 |
| 6 | N-Acetylaspartic acid | ++++ | 4 | 1 |
| 7 | Nevirapine | ++++ | 0.25 | 25 |
| 8 | Tretinoin | ++++ | 0.25 | 1 |
| 9 | Ethynodiol diacetate | +++ | 0.25 | 1 |
| 10 | Pasiniazid | ++ | 2 | 1 |
| 11 | NCS-382 | ++ | 0.25 | 1 |
| 12 | Amcinonide | + | 0.25 | 5 |
| 13 | Thioctic acid | + | 0.5 | 5 |
| 14 | Promazine HCl | + | 4 | 10 |
| 15 | D-Gluconic acid calcium salt | + | 2 | 25 |
| 16 | Benzalkonium chloride | – | 2 | n/a |
| 17 | Chloroacetoxyquinoline | – | 0.125 | n/a |
| 18 | Acyclovir | – | 2 | n/a |
| 19 | Diphenhydramine | – | 0.5 | n/a |
| 20 | Reserpine | – | 2 | n/a |
| 21 | Resorcinol monoacetate | – | 2 | n/a |
| 22 | Khellin | – | 2 | n/a |
| 23 | Butenafine HCl | – | 0.5 | n/a |
| 24 | Phenothrin | – | 0.25 | n/a |
| 25 | Nomifensin maleate | – | 2 | n/a |
| 26 | Ethopropazine HCl | – | 1 | n/a |
| 27 | Fosphenytoin sodium | – | 0.25 | n/a |
| 28 | Thiram | – | 0.5 | n/a |
| 29 | Vitamin K3 | – | 0.25 | n/a |
| 30 | BOC-S-acetaminomethyl-L-cysteine → N-Acetyl-L-cysteine** | – | 0.5 | n/a |
| 31 | RIAA 94 | nd | 0.5 | nd |
| 32 | Trientine | nd | 1 | nd |
| 33 | Beta propiolactone | nd | 2 | nd |
| 34 | Emodic acid | nd | 0.5 | nd |

All 23 Hit II drugs (non-shaded), as well as 2 Hit I validated compounds with high SSMD values (shaded light gray), and 9 Hit I 'fails' (shaded dark gray), are listed. The top 30 drugs were tested for increased β-cell numbers: 15 were validated as leads (++++ = p < 0.0001; +++ = p < 0.001; ++ = p < 0.01; + = p < 0.05), 15 failed (–). *optimal response concentration for the ARQiv and validation screens; **substituted compound due to availability issues; n/a: not applicable; nd: not determined. n = 5–10 larvae per condition, experiment repeated 2–3 times.

following a 2-day exposure (3–5 dpf) to fluoxetine (25 μM) or serotonin (25 μM). Both treatments displayed a significant increase in β-cell numbers (fluoxetine, 42.6 ± 2.0; serotonin, 45.7 ± 2.5; DMSO, 33.7 ± 1.2) suggesting that elevated serotonergic signaling promotes β-cell proliferation (*Figure 6A*).

To verify effects on cell division, we combined SSRI treatments with 5-ethynyl-2′-deoxyuridine (EdU), a thymidine analog that labels proliferating cells (*Salic and Mitchison, 2008*). We used another

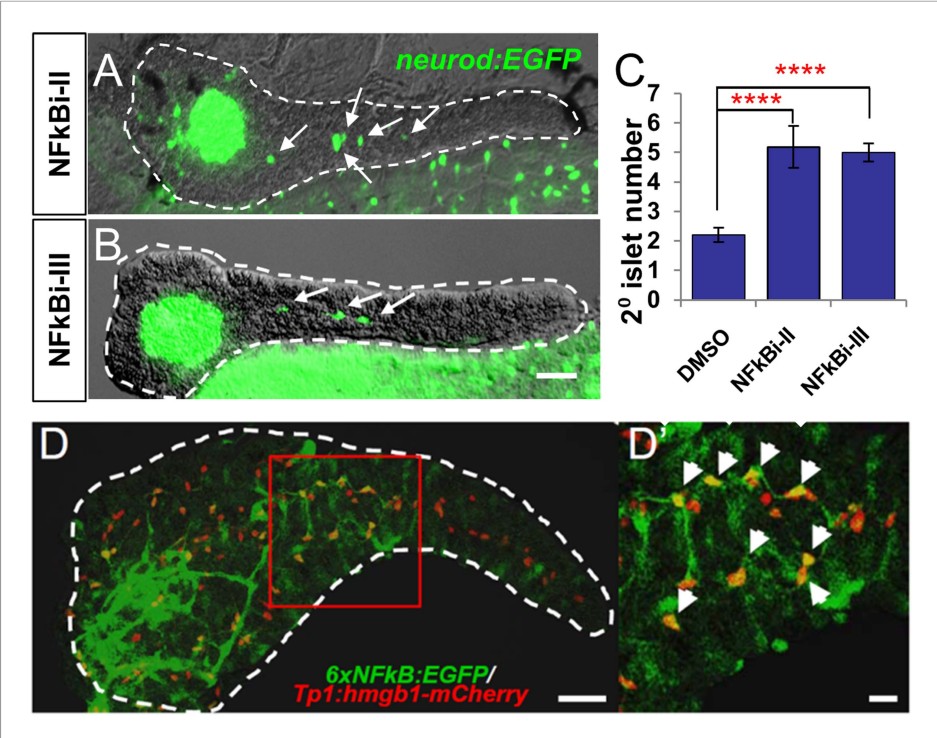

**Figure 5**. NF-κB pathway inhibition induces endocrine differentiation. (**A**, **B**) Representative confocal images—brightfield and fluorescence images merged—of dissected pancreata (dashed lines) from *neurod:EGFP* transgenic larvae treated from 3 to 5 dpf with NF-κB signaling inhibitor II (**A**) or III (**B**). Both inhibitors induced precocious secondary islet formation (white arrows). (**C**) Secondary islet numbers were quantified and plotted relative to vehicle control (0.1% DMSO). n = 5–10 larvae per condition, experiment was repeated 3 times. Error bar, standard error. All p-*values* were calculated using Dunnett's test. ****p < 0.0001. (**D**, **D'**) Representative in vivo confocal z-projection of pancreas (dashed lines) in *6xNFκB:EGFP* ;*Tp1:hmgb1:mCherry* double transgenic larvae at 5 dpf showing co-labeling of the NF-κB reporter (green) and Notch reporter (red) in endocrine progenitor cells (arrows in **D'**), suggesting endocrine progenitors respond to both Notch and NF-κB signaling. Scale bars, 25 µm (**D**), 10 µm (**D'**).

The following figure supplement is available for figure 5:

**Figure supplement 1**. Thioctic acid and parthenolide inhibit NF-κB signaling.

transgenic line labeling β-cell nuclei with GFP, Tg(*ins:hmgb1-EGFP)jh10*, to facilitate quantification of EdU-labeled cells. Larvae were exposed to compounds and EdU for 2 days (3–5 dpf), then fixed and sectioned for imaging. As expected, the results show increased numbers of β cells in principal islets of larvae treated with paroxetine (37.1 $\pm$ 1.2) and RO4929097 (44.1 $\pm$ 2.2), relative to DMSO (29.7 $\pm$ 1.4; *Figure 6B*). Increased numbers of proliferating (EdU$^+$) β cells were also observed for both paroxetine and RO4929097 (*Figure 6B*, EdU$^+$). However, when adjusted for absolute numbers of β cells, only paroxetine-treated larvae showed a significantly higher percentage EdU$^+$ β cells (23.7 $\pm$ 2.7%, vs 11.0 $\pm$ 1.5% and 15.6 $\pm$ 2.1% for DMSO and RO4929097, respectively; *Figure 6C–F*). Combined with our data suggesting paroxetine acts directly on β cells (*Figure 4B–F*), these results strongly suggest that enhanced serotonergic signaling promotes proliferation of β cells.

Serotonin is known to be expressed in human (*Eriksson et al., 2014*) and mouse (*Kim et al., 2010*; *Ohara-Imaizumi et al., 2013*) islets and is implicated in regulating β-cell proliferation during pregnancy (*Kim et al., 2010*) and glucose-stimulated insulin secretion (*Ohara-Imaizumi et al., 2013*). Others have shown that serotonin influences insulin secretion (*Isaac et al., 2013*), and consequently glucose levels. In different model systems, elevated glucose levels have been shown to impact β-cell proliferation (*Porat et al., 2011*) and differentiation (*Maddison and Chen, 2012*). We wanted to

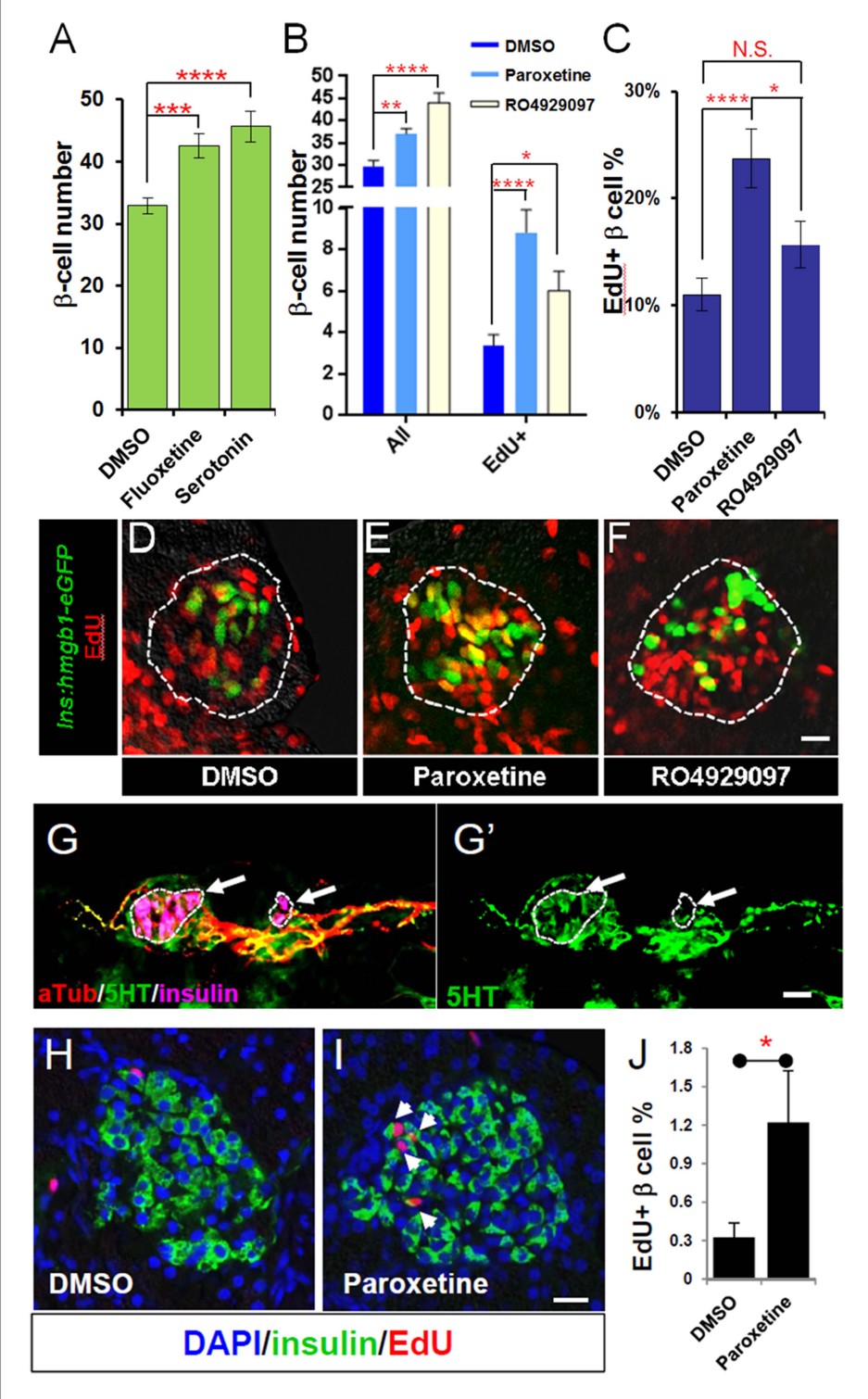

**Figure 6**. Serotonin signaling stimulates β-cell proliferation in a cell type-specific manner. (**A**) β-cell quantification following 25 μM serotonin or 25 μM fluoxetine treatment of *ins:hmgb1-eGFP* transgenic larvae from 3 to 5 dpf indicates enhanced serotonin signaling increases β-cell numbers in zebrafish larvae. (**B**) β-cell quantification in the principal islet (All) and the number of EdU-labeled β cells (EdU+) are plotted following treatments with EdU and either DMSO, 1 μM paroxetine, or 5 μM RO4929097. More β cells overall, and more EdU+ β cells, are observed with 1 μM paroxetine and 5 μM RO4929097 treatments, suggesting effects on β-cell proliferation. (**C**) Plot of EdU+ β cells as a percentage all β cells shows that paroxetine treatment stimulates β-cell proliferation, whereas Notch inhibition

*Figure 6. continued on next page*

*Figure 6. Continued*

does not. Error bars, standard error. (**D**–**F**) Single-plane confocal fluorescence images of *ins:hmgb1-eGFP* islets (dashed lines) treated with EdU and either DMSO (**D**), 1 µM paroxetine (**E**), or 5 µM RO4929097 (**F**)—β cell nuclei (green); EdU$^+$ cells (red); double-labled EdU$^+$ β cells (yellow). Scale bar, 10 µm. n = 5–10 larvae per condition, experiment was repeated 3 times. (**G**, **G′**) Confocal images of immunostained adult zebrafish pancreas indicate that serotonin signaling is active in islets (white arrows, islets indicated by dashed lines). aTub: acetylated tubulin (red); 5HT (5-hydoxytryptamine): serotonin (green); insulin (magenta). Scale bar, 10 µm. (**H**, **I**) Confocal images of adult zebrafish pancreas following injections with EdU and either DMSO (**H**) or 1 mM paroxetine (**I**), and immunostained as indicated. (**J**) Plot of EdU$^+$ β cells as a percentage all β cells shows that paroxetine treatment stimulates β-cell proliferation in adult zebrafish. Error bars, standard deviation. n = 3–5 adult fish per condition, experiment was repeated 3 times. All p-*values* were calculated using Dunnett's test. *p < 0.05, **p < 0.01, ***p < 0.001, ****p < 0.0001.

The following figure supplements are available for figure 6:

**Figure supplement 1**. Amcinonide increases β-cell mass by inducing hyperglycemia.

**Figure supplement 2**. Paroxetine injection stimulates β-cell proliferation in mice.

---

know, therefore, whether serotonin signaling directly affects β-cell proliferation in zebrafish larvae or whether its impact is mediated through increased glucose levels. Using a colorimetric assay to quantify larval glucose levels, paroxetine incubation from 3 to 5 dpf demonstrated no effect on larval glycemia—albeit transient increases on days 1 or 2 of treatment cannot be excluded. Indeed of all leads tested, only the glucocorticoid amcinonide significantly elevate glucose levels (*Figure 6—figure supplement 1*). Thus, we conclude that serotonin promotes increased β-cell numbers independent from affecting glucose levels.

Although serotonin was shown to be present in a subpopulation of cells in the zebrafish intestine, no report has been made regarding its localization in the adult zebrafish pancreas (*Uyttebroek et al., 2010*, *2013*). Using a validated serotonin antibody (*Uyttebroek et al., 2010*), we found this molecule localized to adult pancreatic islets (*Figure 6G,G′*). Immunolabeling was evident along a serotonergic nerve that appeared to innervate the pancreas (indicated by overlap with anti-tubulin staining, that is, 'yellow' regions in merged image, *Figure 6G*) and was also consistent with possible expression within β cells (overlap with insulin, compare *Figure 6G,G′*). The observed expression pattern for serotonin is consistent with immunohistochemistry data from human pancreatic tissue and mammalian model systems. To test if serotonergic signaling also enhances proliferation in the adult zebrafish, we injected mature fish (>3 month old) with a 20 µl mixture of paroxetine (1 mM) and EdU (25 µM) every other day for 10 days (5 injections), followed by an injection of EdU alone on day 12. Fish were sacrificed on day 14 and pancreata sectioned and immunostained for insulin and EdU. The data show a significant increase in EdU$^+$/insulin$^+$ cells in paroxetine-treated fish (3.7-fold, p < 0.05; *Figure 6H–J*). Finally, to check if the effect of paroxetine is conserved in mammals, we performed intraperitoneal injections of paroxetine (15 mg/kg) in young mice daily from the age of postnatal day 6 (P6). EdU intraperitoneal injections (60 µg per injection) were carried out every other day from P8. Pancreata were then collected at P14 and P21 for staining of β cells (Nxk6.1 antibody) and EdU. The data showed a significant increase in β-cell proliferation in paroxetine-treated animals at P14 (DMSO control: 11.2 ± 0.4%, n = 4 animals; paroxetine: 13.8 ± 0.4%, n = 5 animals, p < 0.0001) and P21 (DMSO: 14.8 ± 0.5%, n = 5 animals; paroxetine: 18.06 ± 0.4%, n = 7 animals, p < 0.0001; *Figure 6—figure supplement 2*). Combined with prior observations in pregnant mice (*Kim et al., 2010*), our data are consistent with serotonergic signaling playing an evolutionarily conserved role in regulating β-cell proliferation in the pancreas.

## Discussion

We have established a versatile and sensitive platform for true high-throughput drug discovery in whole-organisms that is applicable to a wide range of in vivo reporter-based assays. We leveraged the high-throughput capacity afforded by ARQiv to reduce false-call rates using qHTS principles—that is, titration-based primary screening (*Inglese et al., 2006*). To our knowledge, this is the first time qHTS has been applied to whole-organism drug discovery utilizing a vertebrate model system. We were

motivated to utilize this approach as our chemical library, the JHDL, was assembled to 'repurpose' existing drugs for new disease targets (*Shim and Liu, 2014*). Thus, a primary goal was to reveal the maximal number of novel therapeutic opportunities afforded by the JHDL, an endpoint which qHTS facilitates by reducing false negatives. Our screen was specifically designed to identify compounds that elevated transgenic *insulin* reporter activity in larval zebrafish. More specifically, we wanted to find drugs that increased pancreatic β-cell mass, thus, possible new therapeutics for ameliorating β-cell paucity in diabetic patients. To create a platform capable of handling high-throughput volumes, we combined ARQiv with a customized robotic workstation (Hudson Robotics) and the COPAS-XL system (Union Biometrica), which we termed 'ARQiv-HTS'. Using this system, automated quantification of YFP reporter activity (i.e., β-cell numbers) in more than a half-million transgenic larvae resulted in the identification of 177 hit candidates. Secondary assays on a subset of prioritized hit candidates validated the majority of lead drugs tested as being able to increase β-cell numbers by induction of endocrine differentiation and/or stimulation β-cell proliferation.

Compared with our prior manual screen (*Rovira et al., 2011*), ARQiv-HTS significantly increased the number of hits implicated (177 vs 62) and leads validated (24 vs 6). This supports the hypothesis that combining automated large-scale assay platforms, such as ARQiv-HTS, with qHTS can add significant value to whole-organism drug discovery without increasing the time required for the primary screen (*Mathias et al., 2012*; *Hasson and Inglese, 2013*; *Rennekamp and Peterson, 2013*). In terms of sensitivity, we estimate ARQiv was able to detect as few as ten additional β cells in the developing pancreas (e.g., from 30 to 40). We validated ~62% of the hit compounds tested as leads (24 of 39, *Figure 2D*; *Tables 1, 2*), a high hit-to-lead validation rate for HTS-based discovery systems (*Hann and Oprea, 2004*). This supports the concept that whole-organism screening can overcome inefficiencies in HTS drug discovery, such as high false positive and lead compound attrition rates (*Giacomotto and Segalat, 2010*; *Mathias et al., 2012*).

In support of our findings regarding candidate drugs, a recent manual screen of 883 compounds by the Stainier lab, implicated three pathways in promoting β-cell replication in larval zebrafish: retinoic acid (RA), glucocorticoids, and serotonin (*Tsuji et al., 2014*). We identified several compounds in those categories as well and implicated an additional 11 mechanisms of action in affecting β-cell biology (*Supplementary files 1, 2*). Importantly, their results support our interpretation of the data presented here regarding serotonergic signaling promoting β-cell replication. In our prior manual screen, we found that inhibition of RA signaling can maintain pancreatic progenitor cells in an undifferentiated state (*Rovira et al., 2011*; *Huang et al., 2014*). Our results, and those of Tsuji et al., have now demonstrated another role for RA (e.g., tretinoin, compound #31, *Figure 4*) in stimulating β-cell proliferation. Both studies also found that glucocorticoids induce β-cell proliferation (e.g., amcinonide, compound #5, *Figure 4*) indirectly by elevating glucose levels (*Figure 6—figure supplement 1*). Importantly, our screen also revealed a broader range of compound categories that potentially enhance β-cell mass (*Supplementary files 1, 2*). Follow-up of these candidates could suggest multiple new mechanisms for increasing β-cell numbers. Such studies will be facilitated by knowledge of implicated mechanisms of action for most of the compounds in the JHDL. As an example, we explored the role of two mechanisms of action that were potentially shared between two lead drug subsets in regulating endocrine differentiation and β-cell proliferation. A third possibility, compounds stimulating a direct increase in insulin expression without changes in β-cell number remain to be evaluated.

The precocious islet assay we developed (*Parsons et al., 2009*; *Rovira et al., 2011*; *Ninov et al., 2012*; *Huang et al., 2014*) was used to confirm effects of Hit I compounds on endocrine differentiation (*Figure 3*, *Figure 3—figure supplements 1, 2*). Among 20 Hit I compounds tested, 11 were validated as Lead I drugs that promoted endocrine differentiation (*Table 1*). It is possible that Lead I compounds stimulated endocrine progenitor proliferation as well, further experimentation will be required to test this. In follow-up studies, we sought to identify common molecular mechanisms of these drugs. Intriguingly, we found that two Lead I compounds, parthenolide and thioctic acid, inhibit the NF-κB pathway (*Ying et al., 2010*; *Ghantous et al., 2013*). We subsequently verified that NF-κB signaling was active in pancreatic progenitors and defined a novel role for NF-κB signaling in regulating pancreatic development; inhibition of the pathway enhances endocrine differentiation. In keeping with this finding, NF-κB positively regulates expression of the pancreatic progenitor marker, SOX9, in human pancreatic cancer stem cells (*Sun et al., 2013*). As we and others have shown, Sox9 is an important transcription factor in the maintenance of pancreatic progenitor cells under regulation of

Notch signaling (*Kopp et al., 2011*; *Manfroid et al., 2012*; *Shih et al., 2012*). Moreover, a recent study shows that proinflammatory cytokines activate the Notch and NF-κB signaling pathways to promote endothelial transdifferentiation to a HSC fate, indicating a requirement for inflammatory regulation of stem cell numbers (*Espin-Palazon et al., 2014*). Taken together, these data suggest a model where NF-κB and Notch signaling maintain transcriptional regulators essential for progenitor maintenance, including *Sox9* and *Hes/Hey* genes, respectively (*Maniati et al., 2011*).

We reasoned that compounds which increased β-cell numbers without concomitant effects on endocrine differentiation—that is, no evidence of precocious secondary islet formation—were acting to promote cell division. Accordingly, quantification of β cells within the principal islet was used to confirm 15 of 30 Hit II compounds as Lead II drugs for stimulating proliferation of β cells (*Figure 4A*). In terms of mechanism of action, two serotonin reuptake inhibitors, paroxetine and amitriptyline (*Sangdee and Franz, 1979*; *Dechant and Clissold, 1991*), were among the 15 leads promoting proliferation (*Table 2*). As noted above, serotonin was also implicated in a recent manual screen for factors promoting β-cell proliferation in zebrafish (*Tsuji et al., 2014*). However, unique to this study, our primary screen data suggested that paroxetine acts in a cell-type selective manner; increasing β-cell number without affecting δ cells (*Figure 4B–H*). Paroxetine is a more SSRI (*Dechant and Clissold, 1991*), suggesting cell-specific proliferative effects may be mediated through serotonin. This intriguing possibility was tested further by assessing the effect of serotonin and another SSRI on β-cell numbers (*Figure 6A*), confirmed by direct assessments of β-cell division using EdU labeling (*Figure 6B–F*), and supported by expression of serotonin in pancreatic islets (*Figure 6G,G′*). These data are in keeping with preferential uptake of serotonin in human β cells in vitro, and in the pancreas of non-human primates and rats (*Eriksson et al., 2014*). We went on to show that paroxetine stimulated β-cell proliferation in both adult fish (*Figure 6H–J*) and neonatal mice (*Figure 6—figure supplement 2*). Serotonin signaling also increases β-cell mass and insulin secretion in pregnant mice (*Kim et al., 2010*; *Ohara-Imaizumi et al., 2013*). It has been known for over a hundred and 50 years that the pancreas is well innervated. Consistent with a role for neuronal signaling in regulating pancreatic biology, we previously found that disruption of sympathetic innervation in mice leads to abnormal islet structure and loss of functional maturation (*Borden et al., 2013*). Collectively, these findings strongly suggest that neurotransmitters may play significant roles in pancreatic development and β-cell proliferation.

Clinical associations between paroxetine and diabetes have been reported but are controversial, with evidence of both beneficial and detrimental effects (*Weber-Hamann et al., 2006*; *Paile-Hyvarinen et al., 2007*; *Knol et al., 2008*; *Derijks et al., 2009*). As roles for serotonin in β-cell function are also inconclusive (*Isaac et al., 2013*; *Ohara-Imaizumi et al., 2013*), further study will be required to clarify whether serotonergic signaling is a viable therapeutic target for diabetic patients. In addition, neuromodulator drugs are known to be highly promiscuous (*Bianchi and Botzolakis, 2010*), therefore, it will be important to test whether other neurotransmitter pathways also affect islet structure and/or β-cell proliferation. This is also emphasized by the fact that neuromodulators make up the largest subcategory among the 131 ARQiv Call compounds, which remain to be further evaluated (>20 compounds, first shaded set in *Supplementary file 2*).

In summary, we present the first full-scale implementation of ARQiv-based whole-organism HTS. Like HCS methodologies (*Pardo-Martin et al., 2010*; *Rihel et al., 2010*; *Sanker et al., 2013*), ARQiv can take advantage of the versatility afforded by a wide array of reporter-based transgenic resources to provide rapid quantitative phenotyping (*Walker et al., 2012*). Thus, both screening strategies can surmount current 'biological validation' bottlenecks in drug discovery (*Mathias et al., 2012*). HCS approaches are extremely powerful, providing multi-dimensional data that can be used to speed hit-to-lead transition times, rapidly gain mechanistic insights (*Rennekamp and Peterson, 2013*), and identify promising new therapeutics (*Zon and Peterson, 2005*). Unlike HCS approaches, ARQiv provides only single-dimension data, quantification of reporter levels—that is, no corresponding images. However, as we have demonstrated here, ARQiv can be coupled to robotics to push the boundaries of throughput for whole-organism drug discovery. Moreover, ARQiv can detect extremely subtle phenotypic changes (e.g., as few as ten additional β cells) using a completely non-subjective methodology (*Walker et al., 2012*). The increase in throughput afforded by ARQiv-HTS has potent practical benefits, supporting HTS 'best practices' such as qHTS (*Inglese et al., 2006*). By screening the JHDL at multiple concentrations, we substantially increased the number of hits implicated, thus, increasing our chances of identifying an optimal drug target for increasing β-cell mass. The majority of

the validated leads, 18 of 24 (75%), are already approved for use in humans, thus, facilitating relatively rapid clinical translation of these findings. Mechanism of action investigations revealed two important signaling pathways affecting β-cell biology: NF-κB was implicated in regulating endocrine differentiation, and serotonergic signaling was shown to selectively stimulate β-cell proliferation. Studies in human pancreatic cells could establish if the function of these pathways is conserved, better define the roles of inflammatory and neurotransmitter modulation in pancreatic biology, and help to determine the degree to which these findings have clinical relevance.

## Materials and methods

### Zebrafish transgenic lines

All studies were carried out in accordance with onsite ACUC protocols. All fish were maintained at 28.5°C with a consistent 14:10 hr light: dark cycle. Transgenic lines used were *Tg(ins:PhiYFP-2A-nsfB, sst2:TagRFP)lmc01* ('β/δ-reporter'; (*Walker et al., 2012*), *Tg(pax6b:GFP)ulg515* (*Delporte et al., 2008*), *Tg(neurod:EGFP)nl1* (*Obholzer et al., 2008*), *Tg(6xNFκB:EGFP)nc1* (*Kanther et al., 2011*), *Tg (ins:hmgb1-EGFP)jh10* (*Parsons et al., 2009*), *Tg(tp1:hmgb1-mCherry)jh11* (*Parsons et al., 2009*).

### Compound library

We screened the JHDL, a collection of 3348 compounds (*Chong et al., 2006b*; *Rovira et al., 2011*). The majority of the compounds in the JHDL are approved for use in humans: 2290 drugs approved for use by the FDA or international counterparts, another 775 drugs at various stages in clinical trials, and 66 rare drug compounds. In some cases, an active pharmaceutical ingredient (i.e., drug) was included in more than one formulation as a separate compound. However, these were tallied as a single drug, giving a total of 3131 drugs included in the JHDL collection.

### ARQiv-HTS

The salient features of the primary screen performed here are described below. Additional details of the robotics-integrated ARQiv system and the methodologies we apply in pursuing whole-organism HTS in zebrafish larvae can be found here: (*White et al., 2015*).

#### Quality control

The 'robust' strictly standardized mean difference (SSMD*) equation was used to assess quality:

$$Z' = 1 - \frac{3(\sigma_p + \sigma_n)}{|\mu_p + \mu_n|},$$

(where $\mu_p$, $\mu_n$, $\sigma_p$, $\sigma_n$ are the sample mean values and sample standard deviations of the positive and negative controls, respectively). This analysis produced a score of 1.67 with log transformed data, consistent with a 'moderate' control of good quality, thus, an HTS-ready assay (*Zhang, 2011*).

#### Sample number calculation

Using positive (DAPT) and negative (DMSO) control data sets generated for quality control tests, we estimated a required sample size using the statistical power calculation:

$$n = \frac{2\sigma^2 (Z_\beta + Z_{\alpha/2})^2}{(\mu_p - \mu_n)^2} \times (1.15),$$

(where, $Z_\alpha$, $Z_\beta$, $\sigma$, $\mu_p$, $\mu_n$ represent the desired level of statistical significance, desired power, standard deviation [of control sample with greatest variance], mean of the positive control, and mean of the negative control, respectively). A power of 99.9% and p = 0.001 for type I (false negative) and type II (false positive) errors—corresponding to a Zα of 3.29 and Zβ of 3.09, respectively—was used to minimize false-call rates. This analysis determined that a sample size of 14 would be sufficient to detect a 50% effect size on β cells (YFP reporter) relative to DAPT positive controls. An R-based code we developed for plotting sample size data is provided as *Source code 1*.

## Hit selection—predicted SSMD score

To estimate a reasonable hit call criterion, 'virtual' assays of the positive (DAPT) and negative (DMSO) control data sets were run. Bootstrapping with replacement was used to run a total of 10,000 computational iterations at a sample size of 16. The following equation was used to predict an SSMD hit score corresponding to a compound producing a 50% effect size relative to the positive control:

$$SSMD = \frac{\Gamma\left(\frac{n-1}{2}\right)}{\Gamma\left(\frac{n-2}{2}\right)} \sqrt{\frac{2}{n-1}} \; \frac{\bar{d}_i}{s_i},$$

(where $\bar{d}_i$, $s_i$, are the sample *mean* and *standard deviation* of $d_{ij}$s—where $d_{ij}$ is the difference between the measured value [usually on the log scale] of the $i$ th compound and the median value of the negative control in the $j$ th plate. $\Gamma()$ is a gamma function). This analysis set an SSMD of $\geq 1.3$ as the hit call cut-off criterion.

## Primary screen (A) drug dilutions and larval dispensing

For the primary screen, larvae were derived from in-crosses of homozygous β/δ-reporter fish (*Walker et al., 2012*). The β/δ-reporter transgene labels pancreatic β cells with yellow fluorescent protein (Phi-YFP, Evrogen) and neighboring δ cells with red fluorescent protein (TagRFP, Evrogen). At the start of each screening session, customized mass fish breeding chambers were used to collect 3000 to 15,000 eggs. At 24 hr post-fertilization (24 hpf), embryos were transferred into a 0.3× Danieu's solution containing 200 nM of 1-Phenyl-2-thiourea (PTU). PTU is a tyrosinase inhibitor, which reduces pigmentation and thereby increases signal-to-noise ratios for ARQiv assays (*Karlsson et al., 2001*). A customized Hudson Robotics system (*Figure 1—figure supplement 1*) was used to dispense and serially dilute individual JHDL stock solutions 1:2 across a 96-well plate (Greiner bio-one, #650209) such that final concentrations were 4 to 0.125 µM in 0.1% DMSO (drug solvent). Positive and negative control 96-well plates were prepared such that they bracketed every 10 drug plates, to account for changes in reporter activity over time (*Figure 1—figure supplement 1C*). Positive control plates consisted of six concentrations of DAPT, a Notch-signaling inhibitor that enhances precocious endocrine differentiation (*Parsons et al., 2009*), serially diluted 1:2 with final concentrations ranging from 200 µM to 6.25 µM. Titrating DAPT in every control plate served to account for lot and/or assay variability over the course of the screen. Negative controls consisted of an entire 96-well plate of drug solvent (0.1% DMSO). After drug and control plates had been prepared, a COPAS-XL unit (Union Biometrica, Holliston, MA) (*Figure 1—figure supplement 1*) was used to sort 3 dpf β/δ-reporter larvae for viability and dispense them into individual wells. All plates were then incubated under standard temperature and light cycle conditions for 4 days until reporter levels were quantified at 7 dpf (*Figure 1E*). 15 min prior to scanning, 10 µl of 0.2% Eugenol (Sigma) in drug solvent was added to anesthetize larvae.

## Primary screen (B) ARQiv scans

Larvae expressing the β/δ-reporter were analyzed using the ARQiv system to quantify *insulin* and *somatostatin 2* reporter levels (YFP and RFP, respectively). Assay parameters were optimized for HTS as detailed in the text. In addition, plate reader detection parameters (e.g., optimal excitation/emission settings, z-dimension 'focus', etc) were determined empirically using previously described methods (*Walker et al., 2012*). Briefly, all wells were scanned in a 3 × 3 grid, with all grid regions analyzed independently. 'Signal' was defined as any region producing a reading greater than or equal to three standard deviations above non-transgenic control fish averages. If more than one region produced 'signal', these values were summed to obtain the total signal for that well. A set 'gain' for the plate reader was established for all scans in order to normalize data across numerous days/scans.

## Real-time HTS data analysis and ARQiv call criterion

We developed MATLAB (*Walker et al., 2012*) and R-based scripts to analyze and graphically present all primary screen data in real-time (for an example, see *Figure 2*). The resultant graphic provided results for each compound in three formats: (1) box plots showed the range of the data at each

concentration; (2) SSMD values provided a measure of the relative strength of potential hits; and (3) a heat map of each plate facilitated initial visual follow-ups for detecting 2° islet formation.

## MATLAB and R-based data analysis

ARQiv data files from the primary screen were saved in an XML format and processed using MATLAB 2008a and/or R. An extraction algorithm was used to determine the total fluorescent signal of each well and tagged with experimental condition information for future analysis (*Walker et al., 2012*). For wells where the signal could not be detected, the maximum regional value was used. A plate parsing algorithm (*Walker et al., 2012*) was used to separate control and drug plates into groups and blocks. Each block consisted of 14 plates including ten drug plates and the flanking sets of positive and negative control plates. The most effective DAPT concentration was parsed out as a subset of 16 per each control plate and used in the drug plate analysis algorithm. Drug plate analysis was performed on each plate in correlation to respective control data. Each drug plate consisted of six concentrations, resulting in eight conditions total for the analysis. Graphical representations of the data were plotted as described above in the text. To ensure accurate SSMD score calculation, outliers (defined as any well having a signal value greater than three standard deviations from the median of the 16 wells comprising that condition) were removed from the calculations. Empirically, we found that outlier data correlated with blank wells, multiple fish per well, floating larvae, and dead larvae. All curves in graphed data were drawn using either a best fit (solid lines) or single polynomial curve fitting function (dashed lines). The original MATLAB code was updated to an R-based code to facilitate improved graphical outputs. The R-based code for processing a series of drug and control plates configured as described above is provided as *Source code 2*.

## Hit test and lead validations

MATLAB (and/or R) data plots were used to identify which conditions produced a SSMD ≥1.3. Using fluorescence stereomicroscopy, researchers then evaluated larvae in the corresponding wells (and in flanking wells) for evidence of enhanced 2° islet formation. Plates showing evidence of increased 2° islet formation were placed in the Hit 1 subset (*Figure 2D*). In addition, ARQiv hit calls having a SSMD ≥1.75 were placed in the Hit II subset (*Figure 2D*). It is important to note that this left 131 ARQiv Call compounds, those having an SSMD of 1.3–1.74 for which no immediate evidence of 2° islets was found; these remain to be further characterized (*Supplementary file 2*).

## Validation assay I (endocrine differentiation)

For endocrine differentiation validation assays, two transgenic lines facilitating independent tests of Hit I compound effects on endocrine development were used; $Tg(pax6b:GFP)^{ulg515}$ (*Delporte et al., 2008*) and Tg(*neurod:EGFP*)$^{nl1}$ (*Obholzer et al., 2008*). Transgenic embryos were plated in 24-well plates with ≥15 embryos per well at 3 dpf in E3 medium. Newly purchased Hit I compounds were diluted from 25 µM to 0.4 µM in a twofold dilution series, added to 24-well plates, and incubated from 3 to 5 dpf. PTU was not required for validation screens and was therefore not used, allowing assessment of lead compounds independent of any potential effects of PTU. As previously described for endocrine differentiation assays (*Huang et al., 2014*), 5 µM of the Notch inhibitor RO4929097 (Selleck Chemicals) (*Luistro et al., 2009*) was used as a positive control. 0.1% DMSO was applied as the negative control. Larvae were fixed at 5 dpf with 4% paraformaldehyde (PFA) at 4°C overnight. Larval pancreata were dissected and imaged using a Zeiss Axiovert200M inverted microscope. GFP positive cells in the pancreatic ductal region other than the principal islet were counted as 2° islets (*Rovira et al., 2011*). All assays were evaluated using one-way ANOVA and p-*values* calculated with a post hoc Dunnett's test. n = 5–10 larvae per condition, and a minimum of three experimental repeats was performed.

## Validation assay II (increased β-cell number)

To validate Hit I and II compound effects in promoting increased β-cell numbers, the *ins:hmgb1-EGFP* (*Parsons et al., 2009*) line was used; this line facilitates β-cell quantification due to nuclear localization of the GFP reporter. The assay was performed similarly to validation I tests except that Hit drugs were tested from 25 µM to 0.2 µM using a 1:5 dilution series. Pancreata were dissected and imaged with a Nikon A1-si Laser Scanning Confocal microscopy under a 20× objective. β cells were counted for all Z planes using ImageJ (NIH) software. All assays were evaluated using one-way ANOVA (Analysis of

Variance) and p-*values* calculated with a post hoc Dunnett's test. n = 5–10 larvae per condition, and a minimum of three experimental repeats was performed.

## Immunofluorescent staining

Fish were fixed either in 4% PFA (5 dpf larvae) or 10% formalin (adults) at 4°C overnight. Pancreata were dissected and immunohistochemistry performed as described previously (*Huang et al., 2014*). Briefly, pancreata were embedded in paraffin and sectioned at 5 μm. Sections were stained with 4', 6-diamidino-2-phenylindole (DAPI) and processed for immunostaining using the following primary antibodies: serotonin (5-HT; 1:100, Rabbit polyclonal, ImmunoStar); acetylated tubulin (aTub; 1:400, Mouse monoclonal, Sigma); green fluorescent protein (GFP; 1:400, Rabbit polyclonal, Life Technologies), GFP (1:400, Mouse monoclonal, Life Technologies), DsRed (1:400, Mouse monoclonal, Clontech); insulin (1:400, Polyclonal Guinea Pig, Dako). Fluorescently conjugated secondary antibodies were diluted 1:400 dilution (Jackson ImmunoResearch Labs). Images were collected using a Nikon A1-si Laser Scanning Confocal microscopy.

## Glucose level assay

Free glucose level was determined in 5-day-old larvae using a glucose assay kit (BioVision). Briefly, 20 larvae were collected by removing the embryonic media and quickly frozen in liquid nitrogen. Frozen larvae were then thawed on ice and grinded thoroughly in 80 μl glucose assay buffer. Then, glucose concentration were measured and calculated following the manufactory instruction (BioVision).

## β-cell proliferation assay

For direct assessment of β-cell proliferation, the Click-iT EdU Alexa Fluor 647 Imaging kit (Life Technologies) was used. For larval studies, drug treated 3 dpf larval were treated with EdU (working concentration 12.5 μM) and fixed at 5 dpf. In adult fish, 25 μM EdU in E3 medium was injected intracoelomically every other day together with paroxetine or 0.1% DMSO control for 10 days and alone on the twelfth day. In both cases, pancreata were dissected out, embedded in paraffin blocks, sectioned and stained as per the manufacturer's instructions. Images were collected with a Nikon A1-si Laser Scanning Confocal microscopy. EdU-positive β cells were counted using the ImageJ (NIH) software. All p-*values* were calculated using Student's t-test as comparisons were made only between a single treatment condition and the control.

## Paroxetine treatment in adult fish

20 μl of 1 mM paroxetine was injected intracoelomically into adult fish (>3 month old) every other day for 10 days. Control fish were injected with same volume of 0.1% DMSO. Fish injected with paroxetine exhibited the reported behavioral effects connected to SSRI treatment (*Wong et al., 2013*). Fish were sacrificed 14 days after the first injection and fixed in 10% formalin for further evaluation.

## Paroxetine treatment in mice

Paroxetine was injected intraperitoneally daily at 15 mg/kg to wild-type mice (129/C57 mixed background) at postnatal day 7 (P7). EdU injections (60 μg per injection) were given intraperitoneally every other day from P8. Mice were sacrificed via $CO_2$ gas (followed by cervical dislocation) and pancreata dissected out. Pancreata were immediately fixed in 4% PFA overnight at 4°C, then transferred to 30% sucrose (in Phosphate buffered saline, PBS) and left rocking overnight at 4°C. Before embedding in Optimal Cutting Temperature compound (OCT), pancreata were equilibrated in 1:1 30% sucrose: OCT overnight. Embedded tissues were sectioned and stained as previously described (*Borden et al., 2013*). The primary antibody, anti-mouse Nkx6.1 (Developmental Studies Hybridoma Bank), was prepared at a dilution of 1:200. Images were collected using a Zeiss LSM 700 confocal microscope.

## Acknowledgements

We are grateful to Dr Meera Saxena and members of the Parsons and Mumm laboratories for critical reading of this manuscript. We thank John F Rawls for providing transgenic lines. The authors also wish to thank Dr Yue J Wang for providing the cover image. This work was supported by the NIDDK, NIH (1RC4DK090816). Other support was as follows: GW and MJP—Juvenile Diabetes Research Foundation (17-2012-408), and the NIH (R01DK080730); FD -MSCRF 2013 Postdoctoral Fellowship; SKR and JSM—Diabetic Complications Consortium, NIDDK, NIH (12GHSU209). JSS—Science and

Technology Development Fund (FDCT) of Macau SAR (FDCT/119/2013/A3); RK—NINDS (R01 NS073751); JOL—FAMRI, ITCR and Prostate Cancer Foundation. The funders had no role in study design, data collection and interpretation, or the decision to submit the work for publication.

## Additional information

### Competing interests

JSM: Acts as a consultant for, Luminomics Inc., a company which uses drug discovery techniques applied in the text. The other authors declare that no competing interests exist.

### Funding

| Funder | Grant reference | Author |
| --- | --- | --- |
| National Institute of Diabetes and Digestive and Kidney Diseases (NIDDK) | 1RC4DK090816 | Jun O Liu, Michael J Parsons, Jeff S Mumm |
| Juvenile Diabetes Research Foundation International (JDRF) | 17-2012-408 | Guangliang Wang, Michael J Parsons |
| Maryland Stem Cell Research Fund | 2013 Postdoctoral Fellowship | Fabien Delaspre |
| Science and Technology Development Fund (STDF) | FDCT/119/2013/A3 | Joong S Shim |
| National Institute of Neurological Disorders and Stroke (NINDS) | R01 NS073751 | Rejji Kuruvilla |
| Prostate Cancer Foundation (PCF) | | Jun O Liu |
| National Cancer Institute (NCI) | | Jun O Liu |
| National Institute of Diabetes and Digestive and Kidney Diseases (NIDDK) | R01DK080730 | Guangliang Wang, Michael J Parsons |
| Diabetic Complications Consortium | 12GHSU209 | Surendra K Rajpurohit, Jeff S Mumm |

The funders had no role in study design, data collection and interpretation, or the decision to submit the work for publication.

### Author contributions

GW, SKR, AC, RK, MJP, JSM, Conception and design, Acquisition of data, Analysis and interpretation of data, Drafting or revising the article; FD, SLW, DTW, Acquisition of data, Analysis and interpretation of data, Drafting or revising the article; R-L, JSS, JOL, Drafting or revising the article, Contributed unpublished essential data or reagents

### Author ORCIDs

Joong S Shim, http://orcid.org/0000-0003-0167-7307

### Ethics

Animal experimentation: This study was performed in strict accordance with the recommendations in the Guide for the Care and Use of Laboratory Animals of the National Institutes of Health. All of the animals were handled according to approved animal care and use committee (ACUC) protocols of Johns Hopkins University and Georgia Regents University.

## Additional files

### Supplementary files

• Supplementary file 1. ARQiv Hit calls. The 46 compounds implicated as Hit Calls following the ARQiv screen and initial visual assessments of enhanced 2° islet formation are listed. Compounds are

ordered according to SSMD value. Hit I and Hit II subsets are indicated by a check mark in the corresponding column. In addition, Hit I compounds that were evaluated in β-cell proliferation assays alongside Hit II compounds are indicated as 'tested'. Also listed are clinical indication, FDA approval status, or naming convention for compounds approved by FDA counterparts in other countries.

• Supplementary file 2. Outstanding Hit calls. The 131 compounds implicated in the primary screen but which remain to be further evaluated are listed here in subgroups according to their implicated pharmacological activity (shaded subsets). Subcategories included: neuromodulators, glucocorticoids, and retinoids—also implicated by Tsuji et al.,—as well as 11 other categories unique to our study. n/a: not applicable.

• Source code 1. R-based code developed for plotting sample size data.

• Source code 2. R-based code for processing a series of drug and control plates configured.

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
