## [Decision Letter]

Thank you for submitting your work entitled “First quantitative high-throughput screen in zebrafish identifies novel pathways for increasing pancreatic β-cell mass” for peer review at *eLife*. Your submission has been evaluated by Fiona Watt (Senior editor), Tanya Whitfield (Reviewing editor), and three reviewers, who had also seen your previous submission. One of the three reviewers, Wenbiao Chen, has agreed to share his identity.

The reviewers have discussed the reviews with one another and the Reviewing editor has drafted this decision to help you prepare a revised submission.

As you will see, all three reviewers find your manuscript much improved, and all agree that the assays you describe demonstrate an important 'proof of principle' technical demonstration for your high throughput ARQiv system. However, all three reviewers also have significant concerns with the manuscript, which must be addressed in any revision. The full reviews are appended below for your reference, but please pay particular attention to the following points:

1) Reviewers 1 and 2 have some remaining questions concerning the underlying biology of the system. Please add extra experimental detail or discussion as requested.

2) Improve the resolution of the figures where possible.

3) The statistical analysis is a concern and needs strengthening. Re-do the statistical analysis with appropriate tests (i.e. ANOVA with post-hoc correction for multiple comparisons, rather than serial *t*-tests).

*Reviewer #1*:

The revision has markedly improved the manuscript. The authors have successfully addressed my concerns except for the first two. Overall, the study is interesting because 1) it is the first true HTS screen performed in a vertebrate organism; and 2) it has identified new regulators of β-cell mass.

*Minor comments*:

1) The authors only partially took care of my concern that the increase of β cells in some cases may be a consequence of impaired insulin signaling. I would have been completely satisfied on this if glucose was measured within the first day of treatment, rather than after 3 days at which compensation may have run its course. A statement acknowledging this possibility should suffice.

2) The authors only partially addressed my concern on their interpretation of proliferation vs differentiation. The authors were correct that proliferation of both β cells and progenitor cells is interesting, but differentiation from replicated progenitors is still differentiation. The data in Figure 6 does not support that β cell replication plays a major role in paroxetine-induced β cell increase. If the new β cells were all from replication, only 3 β cells would have gained from 6 double-positive cells, accounting less than half of the 7 new β cells. The authors should take this into consideration.

*Reviewer #2*:

In the current resubmission, the authors have substantially improved the manuscript and addressed many, although not all, of the concerns. Overall, the data are now presented in a manner that makes it comprehensible to the reader. My remaining questions:

1) The secondary islet assays. It still seems like a major part of their results depend on the increase of secondary islets from the drugs. In their rebuttal, they state that at day 7, the identification of such islets is “not robust” because of the ins/ss reporters, so they turned to the neuroD and pax6 reporters as orthogonal assays. But what I still don't quite understand is the fact that many of the drugs seem to globally increase neurod:GFP expression: in Figure 3, it seems like the overall increase in neurod:GFP is not confined to the pancreas (i.e. just below the clearly shown dotted line delineating the pancreas). I can accept that there are ectopic islets in the pancreas as marked, but it just seems hard to separate this effect out from the more global effects on neuroD expression. Is there any type of histology that could increase confidence that these are truly secondary islets, and not just an artifact of the transgenic reporter system?

2) Figure 5: It is unclear to me why DMSO produces a “predictable increase in signal”; of which reporter? Also, in Figure 5, what reporter are you referring to here? The NFκB-GFP or *hmgb1-mCherry*?

3) Figure 6: The serotonin staining seems extremely widespread throughout the pancreas, and while it overlaps with insulin, there are many areas that do not. How did you validate the serotonin antibody to ensure this has any specificity in your assay?

*Reviewer #3*:

This is a much improved manuscript that is now easier to read and understand. The re-write has made clear how the whole organism HTS was done, including how big of an effect size they expected to be able to detect, how the data was analyzed, and how the screen was performed – none of which was clear in the earlier version. I appreciate the point that the hits are expected to be (hoped to be!) weaker than the positive control for biological reasons and retract my earlier concerns about this.

The “high-throughputness” is most impressive and represents a significant advance. I am also more convinced in this version by the validation of serotonin and NF-κB hits, with much more details and controls (such as validation of NF-κB in the transgenic). This overall improves the manuscript and better highlights these pathways as interesting from biological and future pre-clinical screening viewpoints.

I still have one quibble:

1) I disagree with the interpretation of doing serial *t*-tests as valid without correction. After all, if you did 100 follow up experiments and showed one test at a measly p=0.05, would you call that a real result, when such a result is quite likely by chance? No. In fact, the initial screening does take this into account with false discovery assessments, but the follow-up studies, which you might consider a “mini-screen” do not. This is incongruous.

One could argue that your screen has now given you a prior expectation for a positive result, so subsequent statistical analyses can reflect this, for example, using a one-tailed *t*-test instead of a two-tailed one, since you actually have a prediction for the directionality of the result. Or other Bayesian methods may be appropriate. However, the number of experiments you do must still be accounted for statistically.

I do note that additional follow-up tests for key hits (serotonin/NFκB) mitigates these flaws somewhat, but that doesn't justify *t*-test mis-use.

[…] I do appreciate that how to statistically handle this kind of data in some sense is still up for debate. However, as this paper will be laying the foundation for future true high-throughput screening in the zebrafish, establishing rigorous standards for follow-up validation is of significant importance. May I recommend a nice discussion of these issues in Colquhoun D. (2014) An investigation of the false discovery rate and the misinterpretation of *p*-values. R. Soc. Open sci. 1: 140216. http://dx.doi.org/10.1098/rsos.140216 as a starting point for thinking through these issues?

[Editors’ note: a previous version of this study was rejected after peer review, but the authors submitted for reconsideration. The previous decision letter after peer review is shown below.]

Thank you for choosing to send your work entitled “Quantitative in vivo high-throughput screen: repurposing drugs for increased β-cell mass” for consideration at *eLife*. Your full submission has been evaluated by Fiona Watt (Senior editor), Tanya Whitfield (Reviewing Editor), and three peer reviewers. The decision was reached after discussions between the reviewers. Based on our discussions and the individual reviews below, we regret to inform you that your work will not be considered further for publication in *eLife*.

As you will see, while the reviewers found the work potentially interesting, all three have substantial and overlapping concerns with the manuscript, including the design, analysis and interpretation of the study, and the advances that it makes over existing work.

*Reviewer #1*:

This work describes an application of the ARQiv system to a high throughput identification of chemicals that stimulate β-cell differentiation or proliferation. It uses novel transgenic zebrafish reporters and imaging to generate statistical measurements of compounds that rise above the noise of such assays. Although the application is important and technically impressive, there are substantial concerns about the manner in which the data is interpreted and presented which make me less than enthusiastic about the manuscript in its current form. I will highlight several key issues:

1) Using a ratio of β/δ cells: It is unclear to me precisely what practical function the δ cells serve in the initial screen. I realize the goal was to find things that specifically worked only in β cells, but wouldn't a compound that simply kept β cell intensity constant, yet decreased δ cell intensity, also show up in the screen? Where is this data used, as the biological differences here could be potentially important?

2) The assay they are using for induced β-cell differentiation: Nowhere do the authors show a photo of validation using the original transgenics used in the screen. Only in Figures 3 and 4, using the neuroD reporter, do they show validation. neuroD marks many cell types in this region, and the representative fish they show in Figures 3 and 4 show many, many ectopic GFP^+^ cells outside of the dashed line presumably marking the pancreas. Without the data from the original, how do we know that these drugs do not simply increase insulin-YFP expression in multiple areas of the embryo, which would have scored as a “hit” in the screen? Furthermore, and in line with this concern, how do they know that counting the neuroD positive cells outside of the dotted area does not simply represent abnormal migration/morphology of the endocrine progenitors? For instance, in Figure 3, it really looks like the large GFP islet is diffuse in the parthenolide treated animal, so it is hard to know if this result truly represents enhanced differentiation or simply a migratory effect.

3) The signal/noise ratio calculation: This is very difficult to interpret in its current form. In Figure 1, what exactly do the black/yellow (or red/yellow) bars indicate? And why is there seemingly no dose response relationship?

4) Positive controls: It is odd that in their initial screen, they used the notch inhibitor DAPT (notoriously insoluble) and this seemed to show a specific effect on β cells. Yet, in Figure 5, they use a different notch inhibitor (RO4929097) which now shows essentially no specific effect on β cells but instead an effect on both β and δ cells. This discrepancy makes it hard to know how to interpret the original screen. In fact, if anything, in Figure 5, it looks like RO4929097 causes a decrease in the number of GFP^+^ cells.

5) Baseline stability of the reporters: In the discussion of 5HT section ('Serotonin signaling selectively increases β-cell proliferation’), the DMSO animals had 29 β cells, which went up to 35 β cells with RO4929097 (the positive control). Yet, in the next paragraph, and shown in Figure 6, the DMSO animals now show that the DMSO animals have 33 β cells, nearly the same number seen in their “positive control”. Granted, this further increases with fluoxetine to 42, but these inconsistencies when looking for relatively modest effects is of great concern, if the baseline stability of the assay (i.e. DMSO from experiment to experiment) is not especially stable.

6) The NF-κB reporter: In Figure 4, the authors show a dual color confocal of an NFκB/Notch reporter line, but yet oddly do not show the effect when they add in their hit compounds. While it is interesting that there are a few cells that co-label, this figure does not help us understand how these compounds may be affecting this signaling pathway in the fish.

*Reviewer #2*:

The manuscript by Wang et al., “Quantitative in vivo high-throughput screen: repurposing drugs for increased β-cell mass”, describes an ARQiv-based quantitative HTS screen in zebrafish for identification of drugs that increase β-cell mass. The screen identified drugs that enhance β-cell neogenesis and drugs that promote β-cell replication. Of those the role of NF-κB and serotonergic signaling were validated with additional experiments. Overall, the study is interesting because 1) it is the first true HTS screen performed in a vertebrate organism; and 2) it identified new regulators of β-cell mass. However, there are several issues with regard to the experimental approaches and the interpretations of the results.

Major issues:

1) Although the whole organism screen is powerful for identifying compounds [that increase β-cell mass, the hits may act direct to increase β-cell mass or indirectly by impairing insulin signaling or glucose metabolism, which in turn induces increase of β-cell mass. The latter may be problematic for therapeutic purposes. The long (from 3dpf to 7dpf) drug treatment in the screening protocol makes it difficult to distinguish the 2 possibilities. This should be addressed in the validation step. For example, do the selected hits increase glucose in the fish? Assuming the new β-cells are functional, do treated animals have a larger capacity to maintain glucose homeostasis under diabetogenic conditions?

2) The conclusions on β-cell replication are based on prolonged EdU labeling. The labeling protocol cannot distinguish whether EdU is incorporated prior to or after β-cell differentiation. This is especially the case for the adult studies where a two-week labeling period was used.

3) The evidence for the conclusion that NF-κB signaling in pancreatic progenitor cells regulate endocrine differentiation needs to be strengthened. For example, it is not shown whether the inhibitors decrease EGFP expression in the ductal progenitor cells.

4) The conclusion that neurotransmitter modulators act through the neuronal signaling pathway is too simplistic. The pancreatic β-cells have many neuronal characteristics, including the ability to synthesize and perceive many neurotransmitters. These modulators may act directly on β-cells.

*Reviewer #3*:

Wang et al. describes a high throughput screen in zebrafish for small molecules that can affect pancreatic β-cell proliferation. Given that similar screens have been previously described (59), including by this group (49), the main advance described here is the high automation of the screening assay, with a secondary novel finding being the link between β-cell production and NF-κB signaling. Unfortunately, I found the description of the screening, especially all the computations, statistical calls, and subsequent validations, to be so sloppily described that it is difficult to understand what was actually done. The work does not appear to have been carefully edited, with major editing errors appearing as early as the Introduction – the sentence starting with “OR-4” appears to be an editing note – and this permeates throughout the figures and the text. My impression is that there is not much of an advance over the other zebrafish pancreas screens. It is also unfortunate that none of their hits appear even close to as effective as their positive control, DAPT (e.g. Figure 3, where DAPT is off the charts and most of the 'hits' look like modest effects at best).

First, they lay out some details of the positive control, DAPT, in Figure 1, but there the mysteries already begin. Figure 1 show some box plots, with color-coding that I do not understand, with a curve fit to the data in some way that is not described, and a reference to a “blue-lined range” that doesn't exist. Somehow this data is used to evaluate hits later, but I already don't understand it, and this is not described in the very superficial Methods section, either. Figure 2 shows a glimpse of the computational logic, but none of this is explained. There is some kind of curve fitting, never once mentioned in the text or Methods, Figure 2 shows something called a “Ranked Avgs”, also never described, and Figure 2 shows a map of SSMD values, but how this is calculated, and how the 1.3 cutoff was chosen, is never said. All this detail is important, but I cannot find it anywhere, in the Methods, or supplement or figure legends.

In the next step, they validate their hits, but I still have concerns. First, none get even close to the effect of the positive control. Second, the Methods say this was evaluated by Student's *t*-test, but surely this needs to be corrected for multiple comparisons. Third, I am unclear about how a secondary islet is counted, as the arrows shown in Figure 3—figure supplement 1 seems to point to more like 6-10 islets, but Figure 3 never gets above 6 islets. Are these best cases only? Are some islets more dispersed, so not counted the same? I can't tell.

Finally, they implicate two signaling pathways, NF-κB and serotonin, in β-cell proliferation. The serotonin pathway is already well described, so the novelty of these results rest on the NF-κB story. Unfortunately, this is not terribly convincing-the compounds from the initial screen, thioctic acid and parthenolide, are not exactly NF-κB go-to compounds, and the other compounds they test in Figure 4 are barely described and should be shown as a dose-response, not a single, mysterious dose. Whether these compounds affect NF-κB signaling in zebrafish in these conditions is also unknown, but they could have tested this directly with their NF-κB reporter fish.

---

## [Author Response]

Reviewer #1:

*The revision has markedly improved the manuscript. The authors have successfully addressed my concerns except for the first two. Overall, the study is interesting because 1) it is the first true HTS screen performed in a vertebrate organism; and 2) it has identified new regulators of β-cell mass*.

Minor Comments:

*1) The authors only partially took care of my concern that the increase of β cells in some cases may be a consequence of impaired insulin signaling. I would have been completely satisfied on this if glucose was measured within the first day of treatment, rather than after 3 days at which compensation may have run its course. A statement acknowledging this possibility should suffice*.

The statement below was added to address this issue:

“Using a colorimetric assay to quantify larval glucose levels, paroxetine incubation from 3-5dpf demonstrated no effect on larval glycaemia – albeit transient increases on days one or two of treatment cannot be excluded.”

*2) The authors only partially addressed my concern on their interpretation of proliferation vs differentiation. The authors were correct that proliferation of both β cells and progenitor cells is interesting, but differentiation from replicated progenitors is still differentiation. The data in*
Figure 6
*does not support that β cell replication plays a major role in paroxetine-induced β cell increase. If the new β cells were all from replication, only 3 β cells would have gained from 6 double-positive cells, accounting less than half of the 7 new β cells. The authors should take this into consideration.*

Having read reviewer 1’s new comments, we now see we were misunderstanding the point being made in previous reviews. For this we apologize. We completely agree with this reviewer that differentiation and progenitor cell proliferation are typically linked, and we agree that additional clarification is needed. In particular, it is possible that some of the Lead I compounds we identified as promoting precocious secondary islet formation do so via stimulating progenitor cell proliferation which subsequently leads to the differentiation of new β cells. To clarify that stimulation of progenitor cell proliferation was a possibility among Lead I compounds we have edited the text to read as follows:

“Among 20 Hit I compounds tested, eleven were validated as Lead I drugs that promoted endocrine differentiation (Table 1). It is possible that Lead I compounds stimulated endocrine progenitor proliferation as well, further experimentation will be required to test this. In follow up studies, we sought to identify common molecular mechanisms of these drugs.”

Importantly, the reviewer’s new comments also helped us to reevaluate changes we had made in the revision that mistakenly linked effects on progenitor cell proliferation with Lead II compounds – those showing no effects on differentiation in any of the assays we performed. In the absence of induction of precocious differentiation, any compound promoting progenitor cell proliferation would result simply in more progenitors, not more β cells. We therefore reasoned that increased β cells, in the absence of effects on differentiation, are due to increased proliferation of existing β cells.

For instance, paroxetine induced more fluorescent signal in the primary assay yet in all subsequent follow up studies did not induce differentiation of pancreatic progenitors. This we showed by lack of precocious secondary islets in larvae treated with this drug. Therefore, one likely explanation for increased fluorescent signal is simply more β cells in the pre-existing principal islet. This we showed to be the case by counting β cells in the principal islet .Since we had shown that no differentiation is induced by serotonin, any effect on progenitor proliferation would have no effect on β-cell numbers. Instead, additional progenitors would remain as progenitors. More progenitors can only produce more β-cells if precocious differentiation was also induced. Indeed, as discussed above, since differentiation and proliferation are often coupled, compounds inducing proliferation in progenitors are likely to cause precocious secondary islet formation and be classified as Lead I compounds.

Increased proliferation of pre-existing β cells would account for increased β-cell mass in the absence of differentiation. In the example of paroxetine, we succeeded in showing increased cell-division in pre-existing β-cells – demonstrating proliferation is indeed affected. This is the same result obtained by Tsuji et al. (2015) using FUCCI fish to test the effects of serotonergic signaling on β-cell proliferation directly.

The reviewer has made reference to an apparent discrepancy regarding this data in Figure 6. In Figure 6, we show data for increased β-cell numbers after paroxetine treatment. On average, 7 additional cells are seen per principal islet. By EdU pulse-fix, on average of 9 cells are labeled (versus an average of 3 in the control). This data would be confusing if labeled cells divide only once as 9 EdU+ cells should have only lead to an increase of 4.5 cells. However, if some cells divide twice many possibilities can occur (e.g. 3 cells dividing twice gives 9 labeled cells and an increase of 6). In addition, some new β cells could have arisen from β cells that had already traversed S-phase. Finally, as this was a pulse-fix assay, some of the labeled cells are likely post-S but pre-M phase. Clearly, these possibilities make it difficult to simply add the number of EdU labeled cells and divide by two, let alone to use that approach to determine which types of cell are dividing.

In summary, we are confident that our interpretation of the data is consistent with the results we observed and have made the following changes to ensure the text is in keeping with our position (note, this includes omission of prior changes which we made in error that linked progenitor cell proliferation to ‘validation’ assays for Lead II compounds – i.e., since, effects on progenitor cell proliferation were never assessed it is inappropriate to state that such an effect was validated).

1) Abstract: Further, we discovered novel roles for NF-κB signaling in regulating endocrine differentiation and for serotonergic signaling in selectively stimulating β-cell proliferation.

2) Introduction: By labeling β cells with a fluorescent protein and quantifying changes in fluorescence after exposure to JHDL compounds, many more drugs were identified that induced endocrine differentiation and/or stimulated proliferation of β cells.

3) Introduction: Secondary confirmation screens were designed to determine whether potential hit drugs induced endocrine differentiation (precocious secondary islet formation) or stimulated β cell proliferation (increased β-cell numbers in the absence of effects on differentiation).

4) Subsection “Primary Screen: ARQiv Assay”: The majority of the 46 Hit I and Hit II compounds underwent a series of ’validation assays’ to confirm effects on endocrine differentiation and/or β-cell proliferation.

5) Subsection: Serotonergic signaling stimulates β-cell proliferation.

6) Subsection “Serotonergic signaling stimulates β-cell proliferation”: Combined with our data suggesting paroxetine acts directly on β cells (Figure 4), these results strongly suggest that enhanced serotonergic signaling promotes proliferation of β cells.

7) Discussion: Accordingly, quantification of β cells within the principal islet was used to confirm 15 of 30 Hit II compounds as Lead II drugs for stimulating proliferation of β cells (Figure 4).

8) Figure 4, legend: Validation of increased β cell proliferation: cell counts.

9) Figure 6, legend: More β cells overall, and more EdU+ 1020 β cells, are observed with 1μM paroxetine and 5 μM RO4929097 treatments, suggesting effects on β-cell proliferation. C) Plot of EdU+ 1021 β cells as a percentage all β cells shows that paroxetine treatment stimulates β-cell 1022 proliferation, whereas Notch inhibition does not.

Reviewer #2:

In the current resubmission, the authors have substantially improved the manuscript and addressed many, although not all, of the concerns. Overall, the data are now presented in a manner that makes it comprehensible to the reader. My remaining questions:

*1) The secondary islet assays. It still seems like a major part of their results depend on the increase of secondary islets from the drugs. In their rebuttal, they state that at day 7, the identification of such islets is* “*not robust*” *because of the ins/ss reporters, so they turned to the neuroD and pax6 reporters as orthogonal assays. But what I still don't quite understand is the fact that many of the drugs seem to globally increase* neurod:GFP *expression: in*
Figure 3*, it seems like the overall increase in* neurod:GFP *is not confined to the pancreas (i.e. just below the clearly shown dotted line delineating the pancreas). I can accept that there are ectopic islets in the pancreas as marked, but it just seems hard to separate this effect out from the more global effects on neuroD expression. Is there any type of histology that could increase confidence that these are truly secondary islets, and not just an artifact of the transgenic reporter system?*

NeuroD in the pancreas is a marker of endocrine progenitors in numerous model species, including zebrafish. In addition, the neuroD:GFP line utilized has been used previously in several studies to mark endocrine progenitors (including our previous work, [64]; and that of other groups, Kimmel et al., 2011; Dalgin et al., 2011 & 2015; Matsuda et al., 2013). NeuroD also labels enteroendocrine cells of the gut, and progenitors elsewhere in GI tract, thus the expression outside the pancreas is normal. Depending on section of the image and rotation of the gut, more or less neuroD cells can be seen. The fact that neuroD is normally expressed in cells other than those in the pancreas is one reason we didn’t try using these fish in the initial screen.

*2)*
Figure 5*: It is unclear to me why DMSO produces a* “*predictable increase in signal*”*; of which reporter? Also, in*
Figure 5*, what reporter are you referring to here? The NFκB-GFP or* hmgb1-mCherry*?*

We thank the reviewer for highlighting this confusing statement. The increase in controls is not due to DMSO but is due to normal increase in signal during days 3 to 5 of development. We have changed the text to simply say:

“All compounds induced a significant reduction of NF-κB reporter activity relative to 0.1% DMSO controls”

*3)*
Figure 6*: The serotonin staining seems extremely widespread throughout the pancreas, and while it overlaps with insulin, there are many areas that do not. How did you validate the serotonin antibody to ensure this has any specificity in your assay?*

The 5HT antibody we used, a rabbit polyclonal from ImmunoStar (#20080), was previously validated in zebrafish by Pietsch et al., 2006, [60] & 2013, Won et al., 2012, and Simonson et al., 2013. As serotonin is conserved across species, the same antibody has been used to label the serotonergic system in numerous other species, including mouse, rat, butterfly, zebra finch, crayfish, snail, and even scorpion! Our anti-5HT labeling shows expression in pancreatic islets (dashed circles) and along serotonergic nerve innervating the pancreas (indicated by the overlap with a-tubulin staining to indicate neuronal processes, i.e., ’yellow’ areas in the merged image). The majority of the areas that do not overlap with insulin outside the pancreas are therefore neuronal processes of a serotonergic nerve(s), in keeping with other studies in zebrafish showing labeling of serotonergic nerves in the enteric system ([60]; Simonson et al., 2013), and that this antibody labels neuronal tracts (Won et al., 2012; [61]). We provide an additional figure showing each channel separately which may help to delineate the neural component (Figure 7).

Author response image 1.**DOI:**
http://dx.doi.org/10.7554/eLife.08261.024

We have edited the following statement associated with Figure 6 to clarify the connection between a-tubulin staining and serotonergic innervation of pancreatic islets:

“Using a validated serotonin antibody (60) […] human pancreatic tissue and mammalian model systems.”

Reviewer #3:

This is a much improved manuscript that is now easier to read and understand. The re-write has made clear how the whole organism HTS was done, including how big of an effect size they expected to be able to detect, how the data was analyzed, and how the screen was performed – none of which was clear in the earlier version. I appreciate the point that the hits are expected to be (hoped to be!) weaker than the positive control for biological reasons and retract my earlier concerns about this.

*The* “*high-throughputness*” *is most impressive and represents a significant advance. I am also more convinced in this version by the validation of serotonin and NF-κB hits, with much more details and controls (such as validation of NF-κB in the transgenic). This overall improves the manuscript and better highlights these pathways as interesting from biological and future pre-clinical screening viewpoints.*

I still have one quibble:

*1) I disagree with the interpretation of doing serial* t*-tests as valid without correction. After all, if you did 100 follow up experiments and showed one test at a measly p=0.05, would you call that a real result, when such a result is quite likely by chance? No. In fact, the initial screening does take this into account with false discovery assessments, but the follow-up studies, which you might consider a* “*mini-screen*” *do not. This is incongruous.*

*One could argue that your screen has now given you a prior expectation for a positive result, so subsequent statistical analyses can reflect this, for example, using a one-tailed* t*-test instead of a two-tailed one, since you actually have a prediction for the directionality of the result. Or other Bayesian methods may be appropriate. However, the number of experiments you do must still be accounted for statistically.*

*I do note that additional follow-up tests for key hits (serotonin/NFκB) mitigates these flaws somewhat, but that doesn't justify* t*-test mis-use. […] I do appreciate that how to statistically handle this kind of data in some sense is still up for debate. However, as this paper will be laying the foundation for future true high-throughput screening in the zebrafish, establishing rigorous standards for follow-up validation is of significant importance. May I recommend a nice discussion of these issues in Colquhoun D. (2014) An investigation of the false discovery rate and the misinterpretation of* p*-values. R. Soc. Open sci. 1: 140216.*
*http://dx.doi.org/10.1098/rsos.140216*
*as a starting point for thinking through these issues?*

We thank the reviewer for correcting our mistake. We have reanalyzed all of the figures comparing more than a single group back to the control using a post-hoc correction for multiple comparisons, namely Dunnett’s test, to derive *p*-values. All corresponding figures, figure legends, tables, and methods have been updated accordingly. We are happy to report that although this resulted in slight alterations in the reported *p*-values, all compounds deemed to have achieved statistical significance in the initial report remain so after performing the Dunnett’s test correction for multiple comparisons (see edits in Tables 1 and 2 regarding changes in rank order).

[Editors’ note: the author responses to the previous round of peer review follow.]

Reviewer #1:

This work describes an application of the ARQiv system to a high throughput identification of chemicals that stimulate β-cell differentiation or proliferation. It uses novel transgenic zebrafish reporters and imaging to generate statistical measurements of compounds that rise above the noise of such assays. Although the application is important and technically impressive, there are substantial concerns about the manner in which the data is interpreted and presented which make me less than enthusiastic about the manuscript in its current form. I will highlight several key issues:

1) Using a ratio of β/δ cells: It is unclear to me precisely what practical function the δ cells serve in the initial screen. I realize the goal was to find things that specifically worked only in β cells, but wouldn't a compound that simply kept β cell intensity constant, yet decreased δ cell intensity, also show up in the screen? Where is this data used, as the biological differences here could be potentially important?

The β/δ ratio was not used to define hit compounds. Hits were defined solely based on the YFP SSMD scores calculated by comparing compound-induced YFP signal to vehicle control YFP signal. Thus, the scenario described, where δ -cell numbers decrease but β-cell numbers remain as per controls, would not have been flagged as a hit because the YFP SSMD score would be <1.3, thus below the ’hit‘ selection cut-off. Conversely, the δ -cell SSMD score would likely have been in the negative range. Such a compound would have been dismissed as a non-hit.

As the reviewer describes, the β/δ ratio enabled detection of compounds that preferentially altered β cell numbers (i.e., increased YFP signal relative to vehicle control) without altering δ cells (i.e., no increase in RFP signal relative to vehicle control) – thus potentially acting specifically on β cells. However, again, this was not the metric used to define hits. We agree with the reviewer concerning the potential importance of compounds having differential ’biological effects‘ on β and δ cells. Indeed, this is highlighted in our study in terms of the novelty of our findings regarding serotonergic signaling. We thank the reviewer for pointing out this confusion and have made the following changes to the text.

Modifications: We have rewritten the text to clarify how relative levels of β and δ cell reporters were applied. We have reinforced that ’Hit Calls‘ were based solely on YFP (β cell) SSMD scores and that SSMD scores were calculated independently for each fluorophore relative to the negative control (0.1% DMSO treated fish). We also re-evaluated the methods used to determine DAPT (positive control) performance and found an error: the use of non-transgenic fish as background rather than DMSO treated transgenic controls. We have corrected the text and corresponding figures which now show the proper DAPT / DMSO signal level rather than S:B (Figure 1). This correction shows that RFP signal detection is approximately four-fold less robust than YFP, due to elevated autofluorescent background in the RFP emission range. In turn, this resulted in a de-emphasis of the YFP/RFP ratio in the updated manuscript which will help to clarify how hit plates were identified.

*2) The assay they are using for induced β-cell differentiation: Nowhere do the authors show a photo of validation using the original transgenics used in the screen. Only in*
Figures 3 and 4*, using the neuroD reporter, do they show validation. neuroD marks many cell types in this region, and the representative fish they show in*
Figures 3 and 4
*show many, many ectopic GFP*^*+*^
*cells outside of the dashed line presumably marking the pancreas. Without the data from the original, how do we know that these drugs do not simply increase insulin-YFP expression in multiple areas of the embryo, which would have scored as a* “*hit*” *in the screen? Furthermore, and in line with this concern, how do they know that counting the neuroD positive cells outside of the dotted area does not simply represent abnormal migration/morphology of the endocrine progenitors? For instance, in*
Figure 3*, it really looks like the large GFP islet is diffuse in the parthenolide treated animal, so it is hard to know if this result truly represents enhanced differentiation or simply a migratory effect.*

Replies to each issue raised:

A) Validation in original transgenics: We have added an additional micrograph showing the effects a representative hit compound on secondary islet formation in the β/δ transgenic line used for the primary screen (Figure 2—figure supplement 1). We have also modified the text to clarify the logic behind pursuing ’precocious‘ 2° islet assays to validate compound effects on endocrine differentiation. Specifically, we emphasize more strongly that detection of endocrine differentiation effects during initial visualize assessments of ’hit call plates‘ at day 7 – thus using the original β/δ transgenic line – was not overly robust (only observed in subsets of treated fish in hit call plates). This is likely because insulin and *sst2* expression are only beginning to emerge within 2° islets at ∼day 8, as these promoters mark fully differentiated cell types. Thus, termination of the primary screen at day 7, to some degree, limited our ability to detect endocrine differentiation effects. Finally, we note that the use of alternative (e.g., ’orthogonal‘) assays for secondary confirmation is a common, if not preferred, practice in HTS drug discovery. Therefore, the strategies we have used for hit validation are in keeping with accepted practices.

B) NeuroD is normally expressed in many areas outside of the pancreas, but GFP^+^ cells outside the pancreas were not included in the analysis of secondary islet formation. The dissected pancreas is quite easily delineated (dashed lines, Figure 3 and Figure 3—figure supplement 1 and Figure 3—figure supplement 2). Regarding the question of abnormal migration/morphology of endocrine progenitors, again neuroD:GFP^+^ cells outside the pancreas were not counted. Migration from the gut into the pancreas is extremely unlikely and has never been described in any developmental model system analyzed.

C) Drugs increase insulin-YFP in multiple areas: all potential hit call plates were visually evaluated immediately after the ARQiv scan using standard fluorescence microscopy. Plates showing ’yellow‘ fluorescence in other than the pancreatic region were eliminated from further analysis, as shown in Figure 2 in updated manuscript (2E in the original manuscript); 29 ’fluorescent compounds‘ referred to autofluorescent and/or compounds suspected of ’staining‘ larval fish. We revised the manuscript to emphasize this point. We saw no expression patterns suggestive of ectopic insulin expression. However, compounds that significantly increased insulin expression levels in general should have been flagged as hits, as the reviewer notes. A priori, we do not know if any hit drugs had this effect. We tested for effects on endocrine differentiation and β cell proliferation. We noted in the discussion the possibility that hit calls that failed to be validated as leads for differentiation or proliferation assays may have direct effects on the insulin promoter.

Finally, regarding possible migratory effects as judged by the ’diffuse‘ appearance of the principal islet in the parthenolide treated micrograph. Principal islet morphology is variable. More importantly, all hit call compounds being were flagged due to evidence of increased β cell-associated signal (>YFP). A migratory effect on endocrine progenitors, without increased differentiation, would not be predicted to produce that effect. Combined with the fact that ten other compounds showed similar expression of *neuroD* and *pax6b* reporters within the tail of the pancreas, we feel the most parsimonious interpretation is increased numbers of secondary islets – consistent with prior studies (Wang, Y. et al. 2011, Development, Parsons, M et al. 2009, MOD, Ninov, N et al. 2012, Development).

*3) The signal/noise ratio calculation: This is very difficult to interpret in its current form. In*
Figure 1*, what exactly do the black/yellow (or red/yellow) bars indicate? And why is there seemingly no dose response relationship?*

We apologize for the confusion and we have now modified the figure in question. In its new form, we hope Figure 1 addresses the reviewer‘s concerns. We more clearly label the y-axis as the ratio of DAPT-treated to DMSO-treated signal, and provide an inset legend showing that yellow lines denote YFP-expressing β cells and red lines denote RFP-expressing δ cells. We also present the data as a standard dose-response curve to avoid confusion.

*4) Positive controls: It is odd that in their initial screen, they used the notch inhibitor DAPT (notoriously insoluble) and this seemed to show a specific effect on β cells. Yet, in*
Figure 5*, they use a different notch inhibitor (RO4929097) which now shows essentially no specific effect on β cells but instead an effect on both β and δ cells. This discrepancy makes it hard to know how to interpret the original screen. In fact, if anything, in*
Figure 5*, it looks like RO4929097 causes a decrease in the number of GFP*^*+*^
*cells.*

We failed to adequately address this point in the first version of the manuscript. There is no discrepancy between the results gained from either DAPT or RO4929097. The Notch inhibitor DAPT does not show β cell specific effects and has a well-documented effect on all endocrine cells. The original Figure 1 showed equivalent effects β and δ cells. To avoid confusion, we modified this figure (now Figure 1) as discussed above to better show dose-responsiveness.

DAPT was chosen for the primary screen as it was readily available at significantly lower cost than RO4929097. Moreover, DAPT was the only published compound shown to have effects on endocrine differentiation in larval zebrafish at the time the screen was initiated. In subsequent validation screens we took advantage of the newer compound as it is a more soluble compound. For the validation screens we didn‘t need to use as much compound so price was not so much of an issue. However, both compounds have similar effects in the assays we performed, we now state this explicitly in the text to clarify.

In a new Figure 4 we show a 3D render to show the increase in β-cell mass caused by paroxetine and RO4929097. The previous Figure 5 had been compiled with single z-plane optical sections – admittedly not the best way to allow easy quantification for the reader. We now include an inset of a single z-plane to make it clear there is no co-localization.

*5) Baseline stability of the reporters: In the discussion of 5HT section ('Serotonin signaling selectively increases β-cell proliferation’), the DMSO animals had 29 β cells, which went up to 35 β cells with RO4929097 (the positive control). Yet, in the next paragraph, and shown in*
Figure 6*, the DMSO animals now show that the DMSO animals have 33 β cells, nearly the same number seen in their* “*positive control*”*. Granted, this further increases with fluoxetine to 42, but these inconsistencies when looking for relatively modest effects is of great concern, if the baseline stability of the assay (i.e. DMSO from experiment to experiment) is not especially stable.*

We have extensive experience with this issue, and have found this level of variability to be quite normal when comparing between different strains of fish or across biological repeats. In our work, the average numbers of β cells observed within the principal islet can differ as much as 3-5 cells across strains or per repeat. For this reason, every experiment is properly (internally) controlled and standardized to a DMSO control.

*6) The NF-κB reporter: In*
Figure 4*, the authors show a dual color confocal of an NFκB/Notch reporter line, but yet oddly do not show the effect when they add in their hit compounds. While it is interesting that there are a few cells that co-label, this figure does not help us understand how these compounds may be affecting this signaling pathway in the fish.*

We have added additional data to show that all the NF-κB inhibitors used in this study dramatically reduce fluorescent readout from the NF-κB reporter (Figure 5—figure supplement 1). We feel it is important to establish which cells are undergoing active NF-κB signaling. Using two transgenic reporters, we show NF-κB signaling is in fact occurring in the progenitors (Figure 5) – this observation fits perfectly with NF-κB inhibition causing precocious endocrine differentiation in a cell-autonomous way.

Reviewer #2:

1) Although the whole organism screen is powerful for identifying compounds [that increase β-cell mass, the hits may act direct to increase β-cell mass or indirectly by impairing insulin signaling or glucose metabolism, which in turn induces increase of β-cell mass. The latter may be problematic for therapeutic purposes. The long (from 3dpf to 7dpf) drug treatment in the screening protocol makes it difficult to distinguish the 2 possibilities. This should be addressed in the validation step. For example, do the selected hits increase glucose in the fish? Assuming the new β-cells are functional, do treated animals have a larger capacity to maintain glucose homeostasis under diabetogenic conditions?

The reviewer has brought up a very salient point; namely, that any compound leading to a rise in glucose levels would be expected to cause a reciprocal increase in β-cell number. To address this potential mechanism, effects of lead compounds on glucose levels were investigated and the results included in the revised manuscript (Figure 6—figure supplement 1). Only one of the lead compounds, amcinonide, increased glucose levels. This is to be expected for a glucocorticoid. No other characterized lead compound had significant effects on glucose. We thank the reviewer for this suggestion and feel that the new data add significantly to this work.

2) The conclusions on β-cell replication are based on prolonged EdU labeling. The labeling protocol cannot distinguish whether EdU is incorporated prior to or after β-cell differentiation. This is especially the case for the adult studies where a two-week labeling period was used.

We thank the reviewer for correcting this interpretation of the data. We have revised the manuscript to denote this limitation. Nevertheless, stimulating proliferation of β cells or their progenitors is equally of interest to the field.

3) The evidence for the conclusion that NF-κB signaling in pancreatic progenitor cells regulate endocrine differentiation needs to be strengthened. For example, it is not shown whether the inhibitors decrease EGFP expression in the ductal progenitor cells.

Data regarding compound effects on the NF-κB reporter line are now included in the revised manuscript (Figure 5—figure supplement 1). The results are consistent with the tested NF-κB inhibitors reducing the expression of the NF-κB reporter in the progenitors which then subsequently differentiate.

4) The conclusion that neurotransmitter modulators act through the neuronal signaling pathway is too simplistic. The pancreatic β-cells have many neuronal characteristics, including the ability to synthesize and perceive many neurotransmitters. These modulators may act directly on β-cells.

We agree with the reviewer and have made corresponding changes to the manuscript. We have revised the text to denote this limitation, eliminating discussion of neuronal signaling as the sole source of neurotransmitters.

Reviewer #3:

*Wang et al. describes a high throughput screen in zebrafish for small molecules that can affect pancreatic β-cell proliferation. Given that similar screens have been previously described (*[59]*), including by this group (*[49]*), the main advance described here is the high automation of the screening assay, with a secondary novel finding being the link between β-cell production and NF-κB signaling. Unfortunately, I found the description of the screening, especially all the computations, statistical calls, and subsequent validations, to be so sloppily described that it is difficult to understand what was actually done. The work does not appear to have been carefully edited, with major editing errors appearing as early as the Introduction – the sentence starting with* “*OR-4*” *appears to be an editing note – and this permeates throughout the figures and the text. My impression is that there is not much of an advance over the other zebrafish pancreas screens. It is also unfortunate that none of their hits appear even close to as effective as their positive control, DAPT (e.g.*
Figure 3*, where DAPT is off the charts and most of the 'hits' look like modest effects at best).*

We apologize for the typos and grammatical errors and have carefully revised the manuscript as these issues clearly detracted from the work. We feel the revised manuscript will be of great interest to many researchers as it establishes methods that can be used for multiple whole-organism screening applications. Moreover, the manner of high-throughput screening described here reaches a capacity not seen before for whole-organism assays. The methods described will allow researchers to utilize existing transgenic lines to find new molecular pathways in their areas of interest. Furthermore, compared to previous high-content screens, ARQiv is more cost efficient – a vital attribute these days. Finally, there are a lot of important statistical matters that do need to be addressed and described to account for signal variability that attends in vivo screening whether in cell culture or model organisms. To do so fully in one paper is difficult. To these ends we have expanded the Methods section in the revised manuscript and put additional details in a methodological JoVE article that will be published as a companion to this work.

It is true, and well noted, that Notch-inhibition is the strongest way to induce endocrine differentiation. Clearly, this is because the progenitors use Notch-signaling to remain undifferentiated. This observation has been made by many people in different organisms and multiple different tissue systems. Notch-inhibition, therefore, would make an unlikely therapeutic agent as the effects would be widespread and detrimental. This has been discussed and published several times elsewhere by us and others. Thus, we were actually encouraged to see compounds having partial effects as compared to Notch inhibition. We were in fact trying to identify compounds that modulated other, hopefully more discrete, pathways that act parallel to, or in concordance with, Notch. We have emphasized this in the Discussion of the revised manuscript.

*First, they lay out some details of the positive control, DAPT, in*
Figure 1*, but there the mysteries already begin.*
Figure 1
*show some box plots, with color-coding that I do not understand, with a curve fit to the data in some way that is not described, and a reference to a* “*blue-lined range*” *that doesn't exist. Somehow this data is used to evaluate hits later, but I already don't understand it, and this is not described in the very superficial Methods section, either.*
Figure 2
*shows a glimpse of the computational logic, but none of this is explained. There is some kind of curve fitting, never once mentioned in the text or Methods,*
Figure 2
*shows something called a* “*Ranked Avgs*”*, also never described, and*
Figure 2
*shows a map of SSMD values, but how this is calculated, and how the 1.3 cutoff was chosen, is never said. All this detail is important, but I cannot find it anywhere, in the Methods, or supplement or figure legends.*

We agree with these comments and have made substantial changes. Figure 1 has been changed and simplified. It includes a new graph with the DAPT dose-response plotted and includes details of the curve fir function. As noted for reviewer #1, the comments also spurred us to review the methods for the plot, revealing an error with regard to the proper negative control which has been corrected (see above for details). The updated figure more clearly demonstrates the utility of DAPT as a positive control. In the updated Results and Methods we now discuss in greater detail what SSMD scores are and how they are calculated to flag potential hit compounds. We also discuss how computational bootstrapping of DAPT and DMSO control datasets was used to estimate a hit cutoff of SSMD ≥1.3. At this score, we predict compounds are causing a change in fluorescence equivalent or greater than half the effect caused by the Notch-inhibitor DAPT. We no longer show the ranked averages of the multiple comparisons data, as further review of that MATLAB function found the corresponding plot to be enigmatic, as the reviewer pointed out.

All the curves in the figures were drawn using either best fit (solid lines) or a single polynomial curve fit function (dashed lines). This information is now included in the expanded Methods.

*In the next step, they validate their hits, but I still have concerns. First, none get even close to the effect of the positive control. Second, the Methods say this was evaluated by Student's* t*-test, but surely this needs to be corrected for multiple comparisons. Third, I am unclear about how a secondary islet is counted, as the arrows shown in*
Figure 3—figure supplement 1
*seems to point to more like 6-10 islets, but*
Figure 3
*never gets above 6 islets. Are these best cases only? Are some islets more dispersed, so not counted the same? I can't tell.*

1) Notch-inhibition is indeed the strongest way to induce endocrine differentiation. However, this effect is not clinically relevant as the Notch pathway is essential for the function of multiple organs. Thus, we were not trying to recapitulate the effects seen with Notch-inhibition. Rather, we were looking for alternative mechanisms for increasing β-cell mass without shutting down an essential signaling pathway, either via endocrine induction or stimulation of β-cell proliferation.

2) In each validation experiment we are comparing the average result produced by one condition (i.e. control) to another condition (i.e. drug). We are only ever comparing a single attribute between conditions (e.g., the number of secondary islets), not comparing different attributes across conditions. If the latter were the case, a multiple comparisons test would be warranted to eliminate the possibility that differences between conditions arose due solely to chance, in particular as the number of attributes compared increases. However, this is not the case for any of our validation tests. For this reason, we believe the Student‘s *t*-test to be appropriate.

3) Indeed some of the examples shown were best cases. We are grateful for the reviewer pointing this out. All images are now examples that better reflect the average result.

*Finally, they implicate two signaling pathways, NF-κB and serotonin, in Β-cell proliferation. The serotonin pathway is already well described, so the novelty of these results rest on the NF-κB story. Unfortunately, this is not terribly convincing-the compounds from the initial screen, thioctic acid and parthenolide, are not exactly NF-κB go-to compounds, and the other compounds they test in*
Figure 4
*are barely described and should be shown as a dose-response, not a single, mysterious dose. Whether these compounds affect NF-κB signaling in zebrafish in these conditions is also unknown, but they could have tested this directly with their NF-κB reporter fish.*

It is true that Serotonin signaling has been shown to be essential for β-cell mass increase seen during murine pregnancy. Serotonin has also been implicated in insulin secretion in cells in vitro. In addition, Tsuji et al., recently performed a manual chemical screen in early fish larvae (72-96 hpf) suggesting serotonin signaling can stimulate β-cell proliferation. However, this was not the emphasis of their manuscript; only a single potential hit compound, trazodone, was validated with no further experimentation regarding the role of serotonin. In addition, whereas Tsuji et al., identified only three neuromodulators, this is the largest class of potential hits in our study, implicating twenty-plus neuromodulators. More importantly, only in our work is data presented suggesting that serotonin may act in a cell-type selective manner; increasing proliferation of β cells, or their progenitors, without altering δ cell numbers. As such, we were motivated to thoroughly characterize this pathway and thus our study is a more comprehensive description of the effects of serotonergic signaling in the endocrine system. To further emphasize this point, in the updated manuscript we now go on to show that paroxetine significantly increases β-cell numbers in larval and adult fish, and neonatal mice. Our results therefore stand as novel and should be disseminated in a journal promoting widespread exposure. Furthermore, considering the number of people using paroxetine as a medication, it is of general importance that potential actions of this drug are highlighted.

Regarding the lack of ’go-to‘ compounds. The JHDL contains drugs approved for use in humans which facilitates repurposing for new indications following new screens. However, much is known about potential mechanisms of action for compounds in such libraries and this information can be used to test the relevance of implicated signaling pathways. Accordingly, we noted that among our validated Lead I drugs, two were known to inhibit NF-κB signaling. Accordingly, we tested two other NF-κB inhibitors, which are well known and as mentioned in original text inhibit the NF-κB signaling at different levels of the pathway. For additional clarity we now mention in greater detail the mechanism of action for these additional compounds. All 4 compounds stimulated endocrine differentiation and as mentioned in text each compound targets a separate part of the NF-κB pathway. We believe this result is consistent with NF-κB signaling being involved in pancreatic progenitor maintenance.

A dose-response was performed on all tested compounds, both for the primary screen and for validation assays. We have more strongly emphasized this point in the revised text. However, for clarity, we show results only from optimal concentrations as mock-ups of dose-responses for all compounds were less visually informative. We have listed optimal concentrations for all compounds in each assay.

To address the reviewer‘s concern as to whether the NF-κB inhibitors do indeed inhibit NF-κB signaling in zebrafish, we have added additional material to Figure 5—figure supplement 1. In this figure the fluorescent signal from an NF-κB transgenic reporter line is diminished by all 4 NF-κB inhibitors used in this study. This is shown both by imaging of transgenic pancreata and quantitatively using ARQiv. We have revised the text appropriately to reflect the addition of this new data.